# On taming the effect of transcript level intra-condition count variation during differential expression analysis: A story of dogs, foxes and wolves

**Diana Lobo**[1,2,3]*, **Raquel Linheiro**[1], **Raquel Godinho**[1,2,3], **John Patrick Archer**[1,2]*

**1** CIBIO, Centro de Investigação em Biodiversidade e Recursos Genéticos, *InBIO* Laboratório Associado, Universidade do Porto, Vairão, Portugal, **2** BIOPOLIS, Program in Genomics, Biodiversity and Land Planning, CIBIO, Vairão, Portugal, **3** Departamento de Biologia, Faculdade de Ciências, Universidade do Porto, Porto, Portugal

* diana.lobo@cibio.up.pt (DL); john.archer@cibio.up.pt (JPA)

**Data Availability Statement:** All data is publically available on NCBI (https://www.ncbi.nlm.nih.gov) under the project accession numbers: PRJEB3197 (runs: ERR266355, ERR266386, ERR266395,

## Abstract

The evolution of RNA-seq technologies has yielded datasets of scientific value that are often generated as condition associated biological replicates within expression studies. With expanding data archives opportunity arises to augment replicate numbers when conditions of interest overlap. Despite correction procedures for estimating transcript abundance, a source of ambiguity is transcript level intra-condition count variation; as indicated by disjointed results between analysis tools. We present TVscript, a tool that removes reference-based transcripts associated with intra-condition count variation above specified thresholds and we explore the effects of such variation on differential expression analysis. Initially iterative differential expression analysis involving simulated counts, where levels of intra-condition variation and sets of over represented transcripts are explicitly specified, was performed. Then counts derived from inter- and intra-study data representing brain samples of dogs, wolves and foxes (wolves *vs.* dogs and aggressive *vs.* tame foxes) were used. For simulations, the sensitivity in detecting differentially expressed transcripts increased after removing hyper-variable transcripts, although at levels of intra-condition variation above 5% detection became unreliable. For real data, prior to applying TVscript, ≈20% of the transcripts identified as being differentially expressed were associated with high levels of intra-condition variation, an over representation relative to the reference set. As transcripts harbouring such variation were removed pre-analysis, a discordance from 26 to 40% in the lists of differentially expressed transcripts is observed when compared to those obtained using the non-filtered reference. The removal of transcripts possessing intra-condition variation values within (and above) the 97[th] and 95[th] percentiles, for wolves *vs.* dogs and aggressive *vs.* tame foxes, maximized the sensitivity in detecting differentially expressed transcripts as a result of alterations within gene-wise dispersion estimates. Through analysis of our real data the support for seven genes with potential for being involved with selection for tameness is provided. TVscript is available at: https://sourceforge.net/projects/tvscript/.

ERR266403, ERR266382, ERR266407, ERR266371, ERR266359, ERR266374, ERR266366 and ERR266400), PRJEB4668 (run: ERR351173), PRJNA185055 (runs: SRR636937 and SRR636938), PRJNA78827 (runs: SRR388737, SRR388740, SRR388766, SRR543733,SRR536881,SRR536883) and PRJNA307604 (runs: SRR3084300, SRR3084299, SRR3084298, SRR3084297, SRR3084296, SRR3084295, SRR3084294, SRR3084293, SRR3084292, SRR3084291, SRR3084290, SRR3084289, SRR3084312, SRR3084311, SRR3084310, SRR3084309, SRR3084308, SRR3084307, SRR3084306, SRR3084305, SRR3084304, SRR3084303, SRR3084302 and SRR3084301). Further details are available in S1 Table including: Species, Publication detail (Study), Sample IDs, Project accession and Run accession.

**Funding:** This work was funded by the project NORTE-01-0246-FEDER-000063, supported by Norte Portugal Regional Operational Programme (NORTE2020), under the PORTUGAL 2020 Partnership Agreement, through the European Regional Development Fund (ERDF), and by research funding from the projects under the references PTDC/BIA-EVF/29115/2017, PTDC/BIA-EVF/2460/2014 and POCI-01-0145-FEDER-029115 co-funded by Operational Competitiveness and Internationalization Program, Portugal 2020 and the European Union via the European Regional Development Fund (ERDF) and by National Funds through FCT. DL, RG were supported by FCT (PD/BD/132403/2017 to DL, contract under DL57/2016 to RG) and JA was supported by Funds through FCT under the references POCI-01-0145-FEDER-029115 and PTDC/BIA-EVL/29115/2017. The funders had no role in study design, data collection and analysis, decision to publish, or preparation of the manuscript. FCT and NORTH2020 url's: https://www.fct.pt/ and https://www.norte2020.pt.

**Competing interests:** The authors have declared that no competing interests exist.

## Introduction

Developments in RNA-seq technology have revolutionized transcriptomic studies by allowing for a rapid hi-resolution view of transcript expression [1]. In a typical RNA-seq experiment, transcript expression profiles are estimated for each sample using a metric based upon the number of sequenced reads associated with each transcript within a reference set [2–8]. Condition dependent expression profiles can then be used in order to identify which transcripts are differentially expressed [9, 10]. A challenge arises due to sources of variation within expression profiles that are independent of, or partially overlapping with, the condition of interest [2, 11–13]. The inclusion of biological replicates reduces the effect of such noise [14, 15], and it has been demonstrated that sufficient replicate numbers outweigh sequencing depth in terms of increasing the accuracy within differential expression experiments [14, 16]. In studies not involving highly controlled isolated environments, RNA-seq data from the rapidly growing repertoire of published works can be incorporated [17–19], if data from a matching condition to that being studied is available. This effectively increases the number replicates although variability can be amplified [20, 21].

Differential expression tools compute a statistical significance for each transcript, based upon the abundance estimates within a condition, that reflect the possibility of that transcript being differentially expressed [9, 10, 22]. To reduce the effect of intra-condition variation across biological replicates on the estimation of abundance several methods have been proposed including ALDEx2 [23], EDASeq [24] and PEER [25]. In addition to these, and more generally applied, are the abundance estimation techniques implemented within established differential expression tools such as DESeq2 [9] and EdgeR [10]. However, when methods are compared, relative to the final sets of transcripts identified as being differentially expressed, variable results are observed [15, 23, 26–29]. This is an indication that the problem of intra-condition variation relative to the detection of differentially expressed transcripts using RNA-seq data has not been completely resolved. Furthermore, there is no consensus on the best approach to use [30].

Here we explore the effects that individual transcripts associated with high levels of intra-condition count variation have on the end results of differential expression analysis using the tool DESeq2 [9]; a tool that is well established and that has consistently demonstrated reliability in identifying differentially expressed transcripts [27, 30, 31]. Our aim is to investigate the possibility of whether or not the removal of transcripts, harbouring the highest levels of intra-condition variation, from the reference set used during differential expression analysis can produce sets of differentially expressed transcripts that display an increased level of confidence. The latter being achieved through either: (a) the direct removal of transcripts previously identified as being differentially expressed, but whose expression patterns are ambiguous, or (b) the indirect addition, or removal, of transcripts to, or from, those previously identified as being differentially expressed as a consequence of alterations in p-adj values. The latter being associated with shifts in the distribution of intra-condition variation, following the removal of transcripts harbouring the highest levels of such variation. A by-product of this is the explicit quantification of the level of intra-condition abundance variation present within the final lists of differentially expressed transcripts.

To aid this exploration we present TVscript, a tool for the identification of transcripts above user-specified levels of intra-condition normalized count variation, the latter being strongly associated with transcript abundance estimation [4–8]. As input TVscript requires one file per condition-associated replicate that contains the per transcript read counts obtained following the mapping of reads from the replicate to a common reference set. As output TVscript produces a set of corresponding count files that are absent of transcripts harbouring

normalized intra-condition count variation higher than that associated with a user specified percentile. These updated count files can be subsequently used within the differential expression tool of choice, in our case DESeq2 [9]. Through multiple iterations of differential expression analysis following filtering at varying thresholds and comparisons back to differential expression analysis performed on non-filtered inputs, the effects of transcripts associated with high intra-condition variation, in relation to quantity and consistency of differentially expressed transcripts identified, can be explored.

Using TVscript we first explore the effects of intra-condition per-transcript read count variation through iterative differential expression analysis experiments involving highly controlled simulated count datasets derived from the available dog reference transcriptome [32], and where the exact level of background intra-condition count variation could be specified as well as a subset-set of transcripts to be over represented across replicates (of second conditions used within each iteration). Next, we explored the effects of intra-condition per-transcript read count variation within two distinct case-studies, involving count data obtained following the mapping of intra and inter-study RNA-seq datasets. Within these case-studies differential expression patterns arising from data derived from brain samples of dogs and wolves (inter-study scenario involving frontal cortex, cerebral cortex, prefrontal cortex and frontal lobe) [32–35], as well as tame and aggressive foxes (intra-study scenario involving prefrontal cortex) [36], generated in the scope of domestication experiments are compared at varying thresholds of intra-condition normalized read count variation exclusion.

In addition to exploring the general effects of transcripts harbouring high levels of intra-condition count variation on the outcome of differential expression analysis, we also had an interest in understanding whether or not there were genes commonly up or down regulated within the brain of both forms of domestic canids (dogs and tame foxes), but simultaneously not so within their "wild/aggressive" counter parts (wolves and wild foxes). Such genes are candidates for being associated with tameness. Domestic dogs present marked behaviour differences from wolves, their wild ancestors, due to the evolution of unique social cognitive capabilities [35, 37, 38]. Tame red foxes resulted from deliberated selection against fear and aggression over several generations of cross-breeding [39] and they present several behavioural and pheno-typical traits that resemble those found in dogs [36, 40, 41].

TVscript is open source and code, a quick start guide and test data, are available (under the GNU General Public License) through the SourceForge project page https://sourceforge.net/projects/tvscript/.

## Materials & methods

### RNA-seq datasets

To explore the effects of intra-condition count variation on the detection of differentially expressed transcripts using real data we used both intra and inter-study datasets. At an intra-study level, we combined publicly available RNA-seq data from brain tissue (prefrontal cortex) of 12 tame and 12 aggressive red foxes (S1 Table) generated within the same study [36]. For the inter-study case, we combined multiple publicly available RNA-seq datasets from several dogs and six wolves [32–35], also derived from brain tissue (frontal cortex, cerebral cortex, prefrontal cortex and frontal lobe) (S1 Table). In relation to the latter, dogs 1 to 6 and wolves 1 to 6 were derived from Albert *et al.*, (2012), dog 6 from Roy *et al.*, 2013, dog 7 (two replicates) from Fushan *et al.*, 2015 and dogs 8 and 9 (three replicates each) from Hoeppner *et al.*, (2014) as described within the table. All the samples were downloaded from the National Center for Biotechnology Information (NCBI) and the European Bioinformatics Institute (EMBL-EBI), covering a wide range of ages, both sexes as well as multiple replicates and sequencing

strategies (S1 Table). We selected these specific case studies because, firstly we were interested in evaluating the effects of intra-condition count variation both at intra and inter-study levels and the domestic dog, being a model organism, has an available high-quality reference transcriptome as well as several high-quality RNA-seq datasets generated across different studies; while secondly, we sought to perform a brief exploratory inter-study scan to investigate if the domestication of both dogs and foxes has resulted in the co-expression of a set of common brain genes, relative to their "wild/aggressive" type, since behavioural modifications are considered to have been the first target in domestication [42].

Reads from all samples were mapped to the dog reference transcriptome [32], which contained 26,107 annotated transcripts (Ensembl CanFam3.1, release 92) [43], using Bowtie v.2.3.4.1 [44]. We did not use a splice-site aware mapper, such as Tophat2 [45] or HISAT2 [46], since introns were not expected to be present within the reference transcriptome. Reads were not mapped to the dog genome as our aim was to explore the effects of intra-condition variation on a predefined set of reference transcripts, and not infer novel transcripts from these previously published datasets. We did not use the fox reference transcriptome (available on Ensembl) for the mapping the fox datasets within our overall analysis as we were interested in directly comparing differentially expressed transcripts from the same underlying reference set identified using the fox, dog and wolf datasets. We did however map all of the fox datasets to the fox reference transcriptome in order to confirm that the proportion of reads mapped was similar to that when using the dog reference transcriptome. The *pileup.sh* script from the BBmap package [47] was used to obtain per transcript abundance estimates (measured by the number of mapped reads to the corresponding transcript) in each sample. Read counts from technical replicates of "Dog_8" and "Dog_9" were averaged and merged into one file (S1 Table), while read counts from the two biological replicates of "Dog_7" were treated separately. Finally, to confirm that the final read count numbers were reliable relative to the dog reference transcriptome, all read datasets from foxes, dogs and wolves were re-mapped using the pseudo-mapper kallisto v0.46.1 that implements a rapid and accurate kmer search based strategy for estimating transcript abundance counts [48]. For each dataset an $r^2$ correlation value was calculated describing the linear correlation between the per-transcripts counts obtained following Bowtie2 mapping and the corresponding abundance counts obtained using kallisto.

## Software

TVscript requires as input: (1) multiple files containing per transcript read counts (one per sample), (2) a file containing the lengths of the transcripts that the reads were mapped to, (3) a percentile threshold value for intra-condition variation and (4) a configuration file that indicates the locations all files as well as the condition allocations of the count files in (1). An example configuration file along with further details is available from the SourceForge project page. The steps that TVscript implements to identify transcripts associated with high levels of disjointed read counts are (see Fig 1 for a workflow): i) each input dataset, containing count values from a particular sample, is allocated to either condition A or B, as indicated within the configuration file; ii) counts are normalized by dividing them by the length of the corresponding reference transcript and by the sum of all counts for that sample; (iii) for each reference transcript (t), the absolute pairwise differences between normalized read counts across all samples within condition A are calculated; (iv) the corresponding variances are calculated; (v) steps (iii) and (iv) are repeated for condition B; (vi) variance scores from each condition are placed in ascending order and associated with corresponding percentiles; (vii) reference transcripts are removed (or filtered) if their variance score is above that associated with the user specified percentile threshold; (viii) raw read counts associated with the remaining transcripts

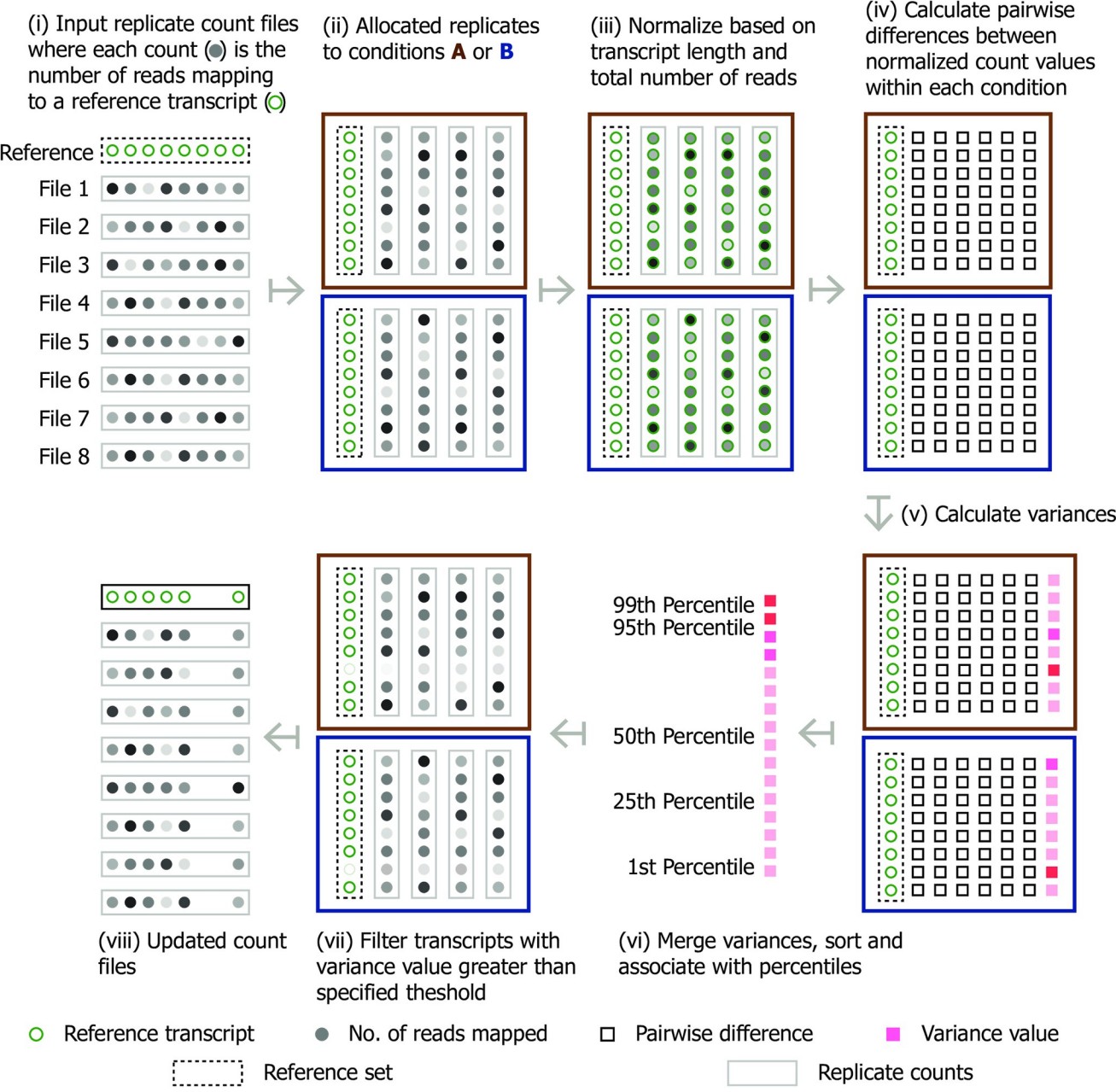

**Fig 1. Diagramatic overview of how reference based transcripts are removed by TVscript.** Steps (i) to (viii) indicate actions taken. Reference transcripts (green circles) are showen for diagramtic purposes in order to highlight how read counts (grey circles) across replicates are treated for each transcript independently. Read counts are grouped into individual files (gray rectangles) in accordance to replicate. These files are grouped into one of two conditions (blue and brown boxes). Remaining keys are indicated at the bottom.

are outputted into separate files that correspond to each input dataset. As to avoid overwriting the original count files, the names of updated count files are specified within the configuration file. These updated count files can be used as input for differential expression analysis software such as DESeq2. Note: in relation to step (vi) the final list of variance scores obtained are a representation of those derived from both conditions, and during step (vii) it is the variance score that is associated with the user-defined percentile that is used. The latter means that for

separate comparisons (e.g. wolves *vs*. dogs and aggressive *vs*. tame foxes) at a given filtering threshold the number of filtered transcripts may not be identical (despite the same reference set being used); as the underlying variance score distribution calculated for each can be different.

TVscript is implemented in Java programming language and runs on all operating systems with installed Java Runtime Environment v.8.0 or higher. It is open source and available under the GNU General Public License v3.0. Source code, usage instructions and sample data can be found on the SourceForge project page: https://sourceforge.net/projects/tvscript/. Although TVscript is implemented in Java the steps involved can be readily implemented within any language (e.g. R or python), using the detailed description provided above as well as the Java source code that is fully available. There are no dependent packages where code is unavailable. At the time of development we choose Java mainly due to its platform independence, which can be an advantage within setting up analysis pipelines involving many different tools. That said we are aware that many differential expression analysis tools are R based and future demand may warrant a supported R version.

## Controlled intra-condition variation within simulated data

We tested TVscript through a series of iterative differential expression analysis experiments involving highly controlled simulated count data. During each iteration the counts associated with each transcript, within each replicate dataset created, represented simulated transcript expression from the dog reference transcriptome and were obtained using CSReadGen [49]. Using the latter, the level of random background count variation away from that required for a normalized even coverage across all transcripts could be specified, as well as a subset of transcripts to be over represented within replicates of a condition. Background count variation refers to varying count levels associated with individual transcripts that are not maintained across replicates of a condition, thus effectively reflecting intra-condition noise, whilst specifying a subset of transcripts to be over represented across replicates of a condition reflects identifiable over expressed transcripts. A similar experiment to those described here, but in relation to the effects of chimerism on the results of differential expression, has been described in Linheiro and Archer (2021) [50].

As a preliminary, and to demonstrate the general reliability of DESeq2 in the absence of random intra-condition count variation, count data was simulated for 22,580 transcripts ranging in length from 300 to 5000 bp from within the dog reference transcriptome. For a single iteration ten replicates of count dataset, each representing three million read pairs ($\approx$20X coverage), were generated in accordance with two conditions, A and B (five in each), where within condition B one hundred transcripts were selected for count over representation by a factor of two across replicates. For all other transcripts, counts were generated to represent an even distribution of reads (length 150 bp, insert size 300 bp). These count files were used as input to perform differential expression analysis between condition A and B in DESeq2 v.1.32.0, considering transcripts with p-adj $<$ 0.05 (corrected by the Benjamini and Hochberg method) to be differentially expressed. The number of transcripts that were detected as being over-expressed was recorded. This was repeated one hundred times in two different ways: (i) the transcripts initially flagged for count over representation were kept constant throughout and (ii) during each iteration a new set of random transcripts for over representation was selected. A brief overview of the R-script we used for differential expression analysis within individual iterations is available on the Zenodo repository [51].

Next, a similar experiment was performed but where the level of random count variation introduced into the count datasets generated ranged from 1% to 10% in steps of one.

Introduced variation was not coupled between replicates, thus reflected intra-condition variation. At each level of variation one hundred iterations of the following steps were performed. (i) Ten replicates were generated and allocated into two conditions A and B (five in each), where within B one hundred selected transcripts had counts over represented. (ii) At the percent level of variation associated with the iteration, that percent of transcripts from each of the ten replicates were randomly selected for count over representation. (iii) DESeq2 was used in a similar manner to before on the count files within conditions A and B to obtain a list of over expressed transcripts. (iv) TVscript was run using a 95th percentile variance threshold to generate ten corresponding modified count files also separated into two conditions (A' and B'). (v) DESeq2 was again used on these to obtain a list of over-expressed transcripts. (vi) The lists of over-expressed transcripts obtained in (iv) and (v) were cross compared. Once again this was repeated in two different ways: (i) the one hundred transcripts initially flagged for over representation were kept constant throughout all levels of variation and for each of the associated iterations and (ii) during each level of variation and for each iteration a new set of one hundred transcripts were randomly selected.

## Exploring the removal of transcripts associated with high intra-condition variation within real data

For each case study (wolves *vs*. dogs and aggressive *vs*. tame fox) we ran TVscript using the count datasets described under the section "RNA-seq datasets" and by applying variance filtering thresholds corresponding to variance values associated with the 70th up to the 90th percentiles (in steps of five), and to the 91st up to the 99th (in steps of one). Steps of one were used in the latter as to allow for transcripts associated with the highest levels of intra-condition variation to be explored in more detail. For each threshold value, only transcripts with variance below that value were maintained. During each run, we recorded the number and IDs of all transcripts that were removed so that they could be cross-compared. Following each run, we used the updated count files produced to perform differential expression analysis using DESeq2 v.1.22.2. Transcripts with p-adj $< 0.05$ (corrected by the Benjamini and Hochberg method) were considered to be differentially expressed. DESeq2 identifies differentially expressed transcripts by estimating gene-wise dispersions and applying shrinking methods to model counts and thus effectively normalize for individual outliers [31]. Distributions of gene-wise dispersions following normalization are conveniently accessible and provide a good metric to visualize the effects of removing transcripts associated differing filter levels. Differential expression analysis using the original non-filtered datasets was also performed. For the aggressive *vs*. tame fox case study batch effects were not considered as all data came from the same study, tissue and sequencing run, additionally no further information about sample preparation was available. For the wolves *vs*. dogs case study we tested for effects based on tissue, primarily for quality control of the final transcripts we drew biological-related conclusions about, and compared results obtained to those in the absence of batch information. In our analysis we used differential expression results based solely on the latter, as firstly, effects associated with tissue at an inter-study level are unpredictable as there are many factors involved, such as precision of dissection, time of dissection, time to dissect, state of individual tissue samples as well as individual who prepared sample, and other than publication or information mentioned for the fox case study, no further information on batches was obtainable. Secondly, although DESeq2 provides an internalized method for accommodating batch effects that we applied (~batch + condition), the results obtained at an intra-study level, with well defined batches, between alternative methods of testing are variable [52]. Lastly, we were primarily exploring the effects of removing hyper variable transcripts on the mechanics of detecting differentially

expressed transcripts and our simulations and case studies were a means to an end in achieving this. As long as input counts for a given filtering threshold within a given case study or a iteration were consistent with those of the initial input data, the effects of removing hyper variable transcripts could be observed, independent of other factors affecting the data prior to analysis. To visualize the overall effects of covariates broadly affecting the relationships between datasets within each case study, we performed a principal component analysis (PCA) using the *plotPCA* function from DESeq2 with non-filtered normalized count data.

To evaluate TVscript we used three metrics that when combined quantify the overall impact of intra-condition variation on downstream differential expression analysis. The metrics were: i) number of ambiguous positives within transcripts identified as being differentially expressed in the non-filtered datasets; ii) distributions of dispersion estimates and outliers in differential expression analysis for non-filtered and all filtered datasets; and iii) discordance in the list of differentially expressed transcripts between non-filtered and filtered datasets (selected percentiles: 97th, 95th, 90th).

**(i) Ambiguous positives.**   We identified transcripts appearing as being differentially expressed when using the non-filtered datasets as input to DESeq2 that were associated with the top 10 levels of intra-condition variation (above the 90th percentile threshold value). These we designated as ambiguous positives. Small numbers of these, relative to the overall number of identified differentially expressed transcripts would indicate that TVScript is having little direct effect on lists of identified differentially expressed genes.

**(ii) Distributions of dispersion estimates.**   For the non-filtered and all filtered input datasets (70th up to the 90th percentiles in steps of five and to the 91st up to the 99th in steps of one) we calculated correlation coefficients ($r^2$) using a linear regression analysis in R [53], between dispersion estimates and the mean of normalized counts, both the latter calculated by DESeq2 during differential expression analysis. Dispersion is inversely related to the mean, as lower mean counts are affected by variation to a higher degree. If a stronger correlation is seen for the filtered input datasets, then this would suggest that the distribution used to model differential expression could be more reliable in relation to identifying differentially expressed transcripts. In addition to this, we retrieved the number of outliers detected by DESeq2, expecting a decrease after each filtering step. Outliers are recognized by the DESeq2 as the points with extremely high dispersion values that cannot by shrunk towards the fit curve. This was performed independently for both case studies.

**(iii) Discordance lists of differentially expressed transcripts between applied filter levels.**   We calculated the proportion of discordance between lists of differentially expressed transcripts produced when using non-filtered and filtered datasets at the 97th, 95th and 90th percentile threshold values. Two types of observed discordances relative to the non-filtered list were considered: (a) transcripts that were lost directly due to filtering or indirectly due to p-adj values no longer being significant, and (b) transcripts that were added due to alterations in p-adj values. Quantifying the nature of these discordances provides insight into the general consistency of genes identified as being differentially expressed across varying filter thresholds. To visualize the overlap between the non-filtered and filtered lists we used the *VennDiagram* v.1.6.20 package in R.

## Gene annotation and gene family analysis

For each case study, differentially expressed transcripts obtained using the non-filtered and filtered datasets were matched to the correspondent gene ID. This was done with the R package BioMart [54] using the Ensembl Gene database (version 94). To begin to identify gene families that displayed similar regulation in both dogs and tame foxes, i.e. relative to the "wild/

aggressive" type, we grouped up and down regulated genes into gene families. Genes within these families were then classified according to whether they were unique to dogs or tame foxes or shared between the two. Within each case study, a gene family was only considered if all the associated genes agreed in relation to their direction of expression (up or down regulation).

## Results

### Mapping success of RNA-seq data

Mapping of the 44 datasets corresponding to dogs and wolves against the dog reference transcriptome revealed an average success of 60% and 58% respectively, in terms of the number of mapped reads (S1 Fig). Similar values between wolves and dogs were expected, given their recent divergence of ~23,000 years ago [55]. Comparable proportions of reads failing to map (~40%) have been previously reported for dog brain samples [33] and are most likely associated with i) novel genes; ii) regions that are not translated despite being transcribed; iii) contamination with genomic DNA; and iv) uncharacterized chimeras and other artefacts within reference sets resulting from library preparation during sequencing [56] and various assembly errors [57]. When a different mapping approach was used for each RNA-seq dataset (i.e. kallisto) transcript abundance counts remained consistent with those obtained following Bowtie2 mapping, as indicated by high $r^2$ correlation values (S2 Fig). $R^2$ values ranged between 0.8546 and 0.9944. All per-transcript mapped read counts, obtained following each mapping approach, have been made available on the Zenodo repository [58]. For the fox datasets, an average of 50% of reads mapped to the dog reference transcriptome using Bowtie2 (S1 Fig); confirmed by the kallisto estimated abundance counts (S2 Fig). This lower percentage of mapped reads, relative to the dog and wolf datasets, could be expected due to an increased genetic divergence from dogs (~10 mya) [59] together with the other aforementioned factors. However, when these fox datasets were mapped to the Ensembl available fox reference transcriptome using both Bowtie2 an improvement in the overall mapped read counts was not observed (S3 Fig).

### Controlled intra-condition variation within simulated data

When not faced with increasing levels of random intra-condition count variation DESeq2 performed exceptionally well. For 86 of the one hundred iterations performed DESeq2 recovered all transcripts that were selected for count over representation (S4a Fig). Of the other 14 iterations the lowest number recovered was 72. Similarly, for the one hundred iterations where the random transcripts selected for count over representation were re-selected during each, in 87 cases all over represented transcripts were identified as being over expressed, whilst in the remaining 13 the minimum number identified was 72 (S4b Fig). However, as levels of introduced intra-condition variation increased, the number of transcripts identified by DESeq2 fell, both before and after filtering the input counts with TVscript (Fig 2). At all levels of intra-condition variation, the post-filtered data had an increase in the number of differentially expressed transcripts identified. It should be emphasized that the reduction in the number of transcripts identified as being over expressed is not a negative reflection on the performance of DESeq2, but instead it is a consequence of purposefully increasing the level of randomness within the count data. The same pattern is true when the one hundred selected transcripts for count over representation are re-selected within each iteration (S5 Fig).

For iterations associated with each increment in random intra-condition variation, the number of transcripts commonly identified as being over expressed both prior-to and post filtering are presented in S2 Table (over represented transcripts kept constant across iterations)

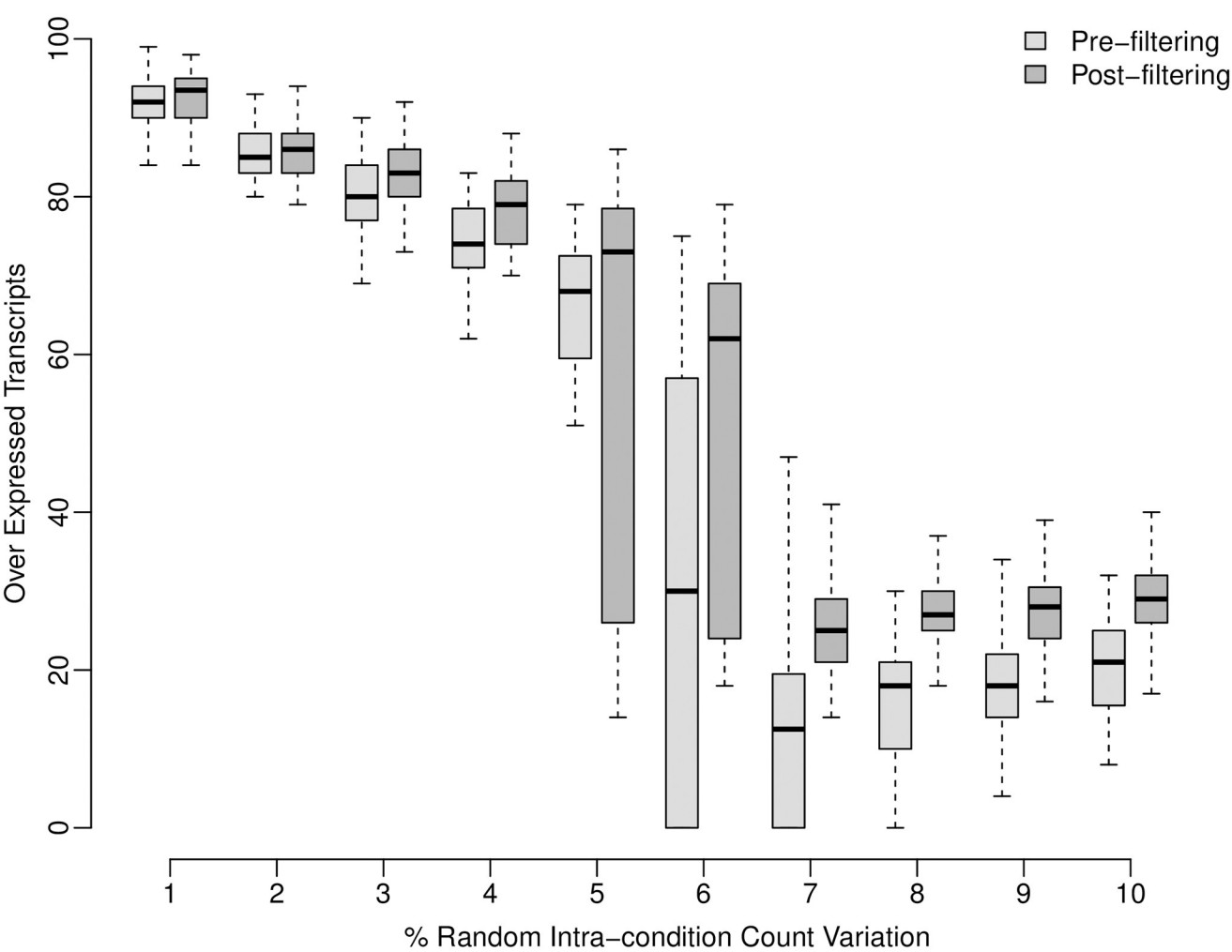

**Fig 2. Over expressed transcripts pre- and post-filtering using simulated data.** The number of transcripts identified by DESeq2 as being over expressed both prior to (light gray) and post (dark gray) filtering of count datasets within each of the one hundred iterations performed at each level of introduced random intra-condition count variation. Each iteration involved initially simulating ten count datasets divided into conditions A and B following which DESeq2 was run to attempt to identify the one hundred transcripts selected for over representation as described in the methods. Following this the ten simulated datasets were filtered using TVscript with a 95th percentile threshold to generate corresponding filtered datasets (divided into corresponding conditions A' and B') on which DESeq2 was re-run.

and S3 Table (over represented transcripts re-selected for each iteration). The proportion that these numbers make up relative to the maximum number of transcripts identified as being differentially expressed, pre- and post filtering, are presented in S4 and S5 Tables (constant) (re-selected). In all cases, below a 5% level of random variation these numbers are high (constant —1 to 4% averages: 0.96, 0.98, 0.94 and 0.81; re-selected—1 to 4% averages: 0.96, 0.97, 0.93 and 0.82), indicating that on top of additional transcripts identified post-filtering with TVscript, transcripts identified pre-filtering are still found. Consequently, this suggests that additional transcripts identified as being over expressed post-filtering are not at the expense of previously detected transcripts pre-filtering. Above the 5% level of intra-condition variation the ability to successfully identify the one hundred transcripts selected for over representation within condition B diminishes within iterations (Fig 2; S5 Fig). This could be indicative of a tentative estimate on the limit of at what level of random intra-condition count variation becomes inhibitory within differential expression analysis studies.

## Exploring the removal of transcripts associated with high intra-condition variation within real data

No significant difference existed between the overall distributions of the per-transcript intra-condition variation values for wolf and dog samples (Wilcoxon-test, p-value < 0.198, Fig 3a). The PCA based on the entire set of normalized non-filtered counts, revealed that the wolf samples were more aggregated than dog samples (Fig 3c). For aggressive and tame fox samples, we observed a significant difference (Wilcoxon-test, p-value $< 2.2e^{-16}$, Fig 3b) between the distributions of the per-transcript intra-condition variation values, most likely resulting from an increased intra-condition variability within tame fox samples. In particular, we found five samples that were differentiated from the remaining seven in the PCA (Fig 3d), with 80% variance being explained by this clustering in PC1.

Prior to differential expression analysis, for each case study (wolves *vs.* dogs and aggressive *vs.* tame foxes), TVscript was used to remove transcripts in accordance with a series of intra-condition variance thresholds (Fig 4a and 4b; S6 Table). Initially, for wolves and dogs 184 transcripts (out of the 26,107) associated with high intra-condition variation (99th percentile and

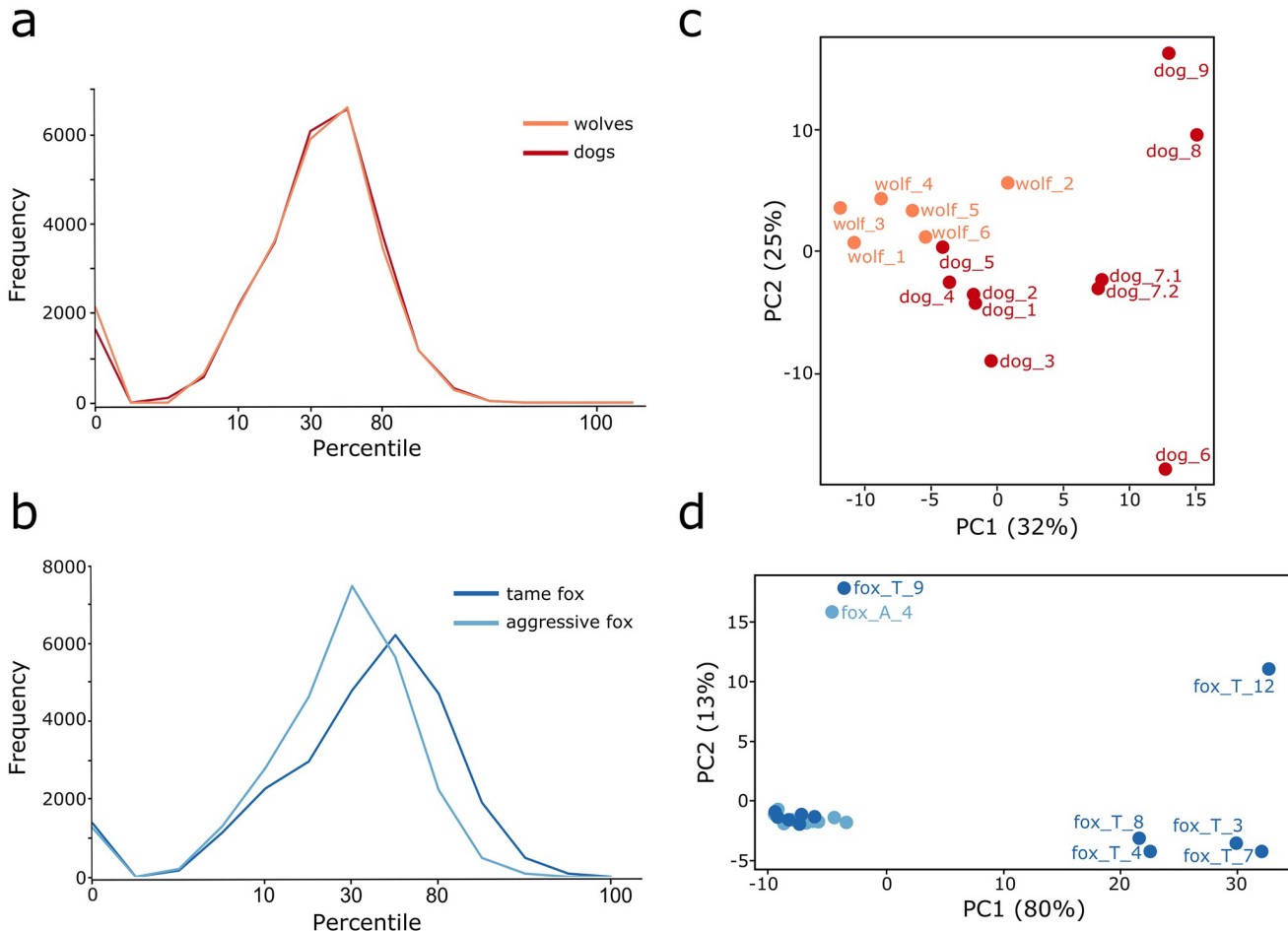

**Fig 3. Characterization of intra-condition variation.** Percentile range of intra-condition variation scores (x-axis) observed prior to filtering, across both case studies, a) wolves (orange) and dogs (red); b) tame (dark blue) and aggressive (light blue) foxes. PCA plots based on normalized non-filtered count data of the individual datasets comparing c) wolf and dog, and d) tame and aggressive fox. In the latter only individual samples that were positioned within a distant cluster are labelled with the sample ID.

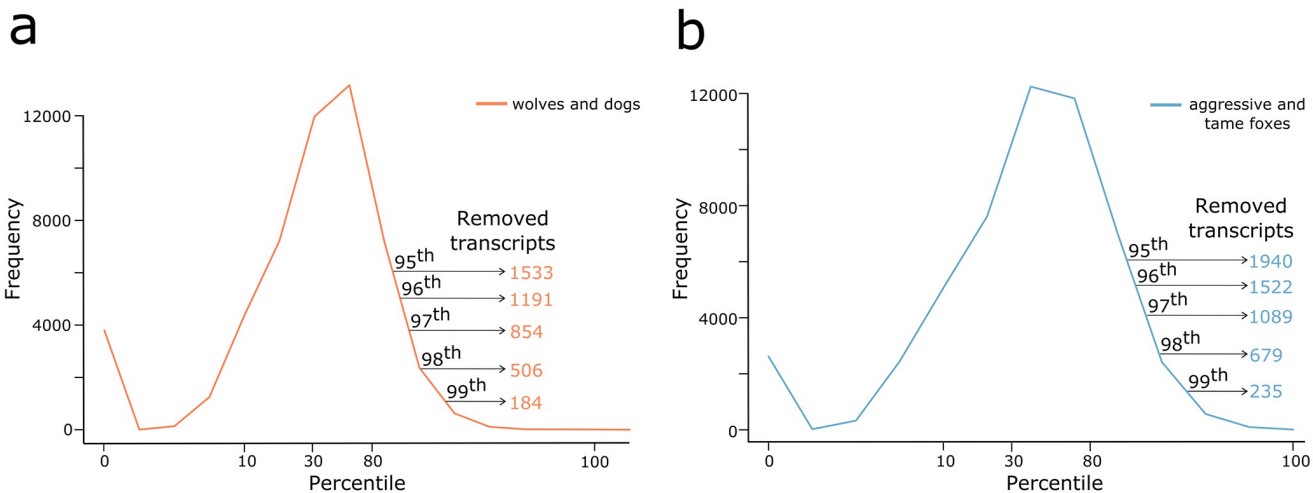

**Fig 4. Removal of transcripts above specified levels of intra-condition variation.** Percentile range of combined intra-condition variation scores (x-axis) present in each case study, a) wolves and dogs; b) tame and aggressive foxes. The number of transcripts removed in the top five percentiles (from the 95th to the 99th) are presented in each panel.

above) were removed, while for the aggressive and tame fox samples, 235 transcripts were removed. The number of transcripts removed was higher for the fox samples than for those of wolf and dog, reflecting the higher intra-condition variability present. Combined across the top ten levels of intra-condition variation, 12% (n = 3134) and 14.89% (n = 3888) of the reference transcripts were removed in wolf/dog datasets and aggressive/tame fox datasets respectively (S6 Table).

## Differential expression analysis

Using non-filtered datasets as input to DESeq2, 430 differentially expressed transcripts were identified between wolves and dogs (Fig 5a; S7 Table). Of those, 259 were up regulated, while 171 were down regulated in dogs. Between aggressive and tame foxes, 651 differentially expressed transcripts were identified (Fig 5a; S8 Table), of which, 532 and 119 were up and down regulated, respectively, in tame foxes. Post filtering, within the first ten steps of size one from the 99th down to the 90th percentiles, the number of differentially expressed transcripts identified peaks at the 97th (n = 430; up = 255, down = 175) and the 95th percentiles (n = 730; up = 607, down = 123) in dogs and tame foxes (Fig 5a), respectively. This indicates that for these data the removal of the 3% (n = 854) and 5% (n = 1940) of transcripts associated with the highest levels of intra-condition variation maximized the detection of differentially expressed transcripts.

## Evaluation metrics

**(i) Ambiguous positives.** Of the transcripts that appeared as being differentially expressed, when using non-filtered datasets as input to DESeq2, 17.44 (n = 75) and 21.51% (n = 140) were associated with high intra-condition variation (above the 90th percentile threshold) within the wolves *vs.* dogs and aggressive *vs.* tame foxes respectively (Fig 5b). This was higher than the relative proportion of such transcripts with the reference set in general, where 12.08% and 14.89% of transcripts possessed intra-condition variation above the 90th percentile for the wolf/dog and the aggressive/tame fox respectively (Fig 4a and 4b; S6 Table). These were transcripts that we considered as ambiguous positives and the average across both case studies

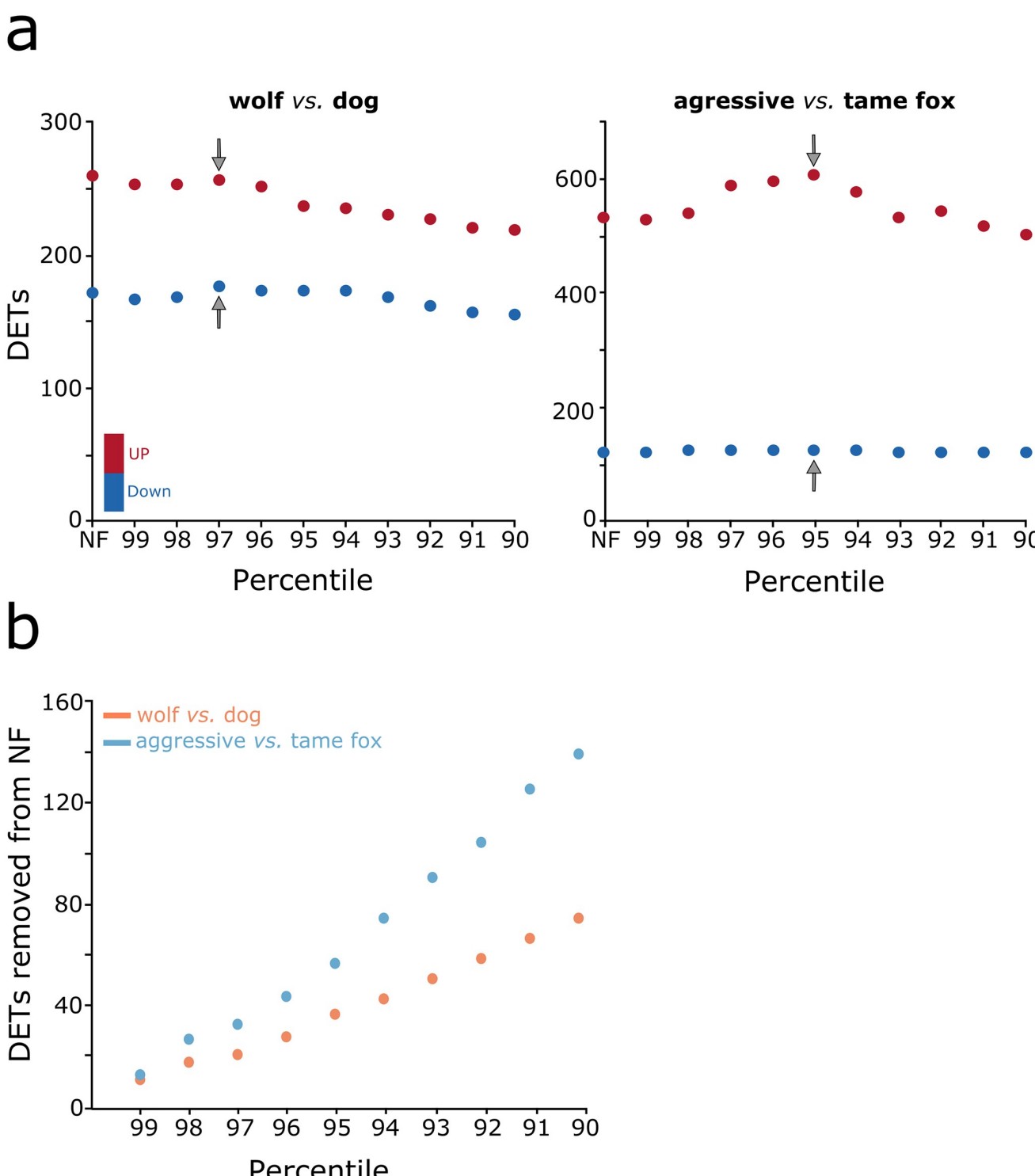

**Fig 5. Effects of removing transcripts above specified levels of intra-condition variation on differential expression analysis.** a) Number of differentially expressed transcripts (DETs) identified using non-filtered (NF) and filtered datasets, based on the top 10 percentiles (99th to the 90th), for both case studies. Up and down regulated transcripts are represented by red and blue dots respectively. Gray arrows identify the selected thresholds for which the results of subsequent corresponding differential expression analysis were used for the identification of candidate transcript associated with tameness within each case study. b) Number of transcripts identified as differentially expressed within the non-filtered datasets that were associated the highest levels of intra-condition variation (99th to 90th) within both case studies, wolves and dogs (orange dots), and tame and aggressive foxes (blue dots).

was 19.45%. The number was higher within the fox datasets where elevated variability among samples was observed, suggesting that differences within intra-condition read counts could have influenced the final outcome of identified differentially expressed transcripts.

**(ii) Distributions of dispersion estimates.** Within both case studies the regression analysis indicates that removing transcripts associated with high levels of intra-condition variation improved correlation coefficients in relation to those from the non-filtered datasets (Fig 6a and 6b; S9 Table). Associated with the elevated levels of variation observed within the fox datasets, there was a better fit within the wolves *vs.* dogs comparison ($r^2 > 0.7$) than that of the aggressive *vs.* tame fox one ($r^2 > 0.5$). For the latter, there was visible elevation in the number of dispersed points around the line of best fit. With the removal of transcripts associated with

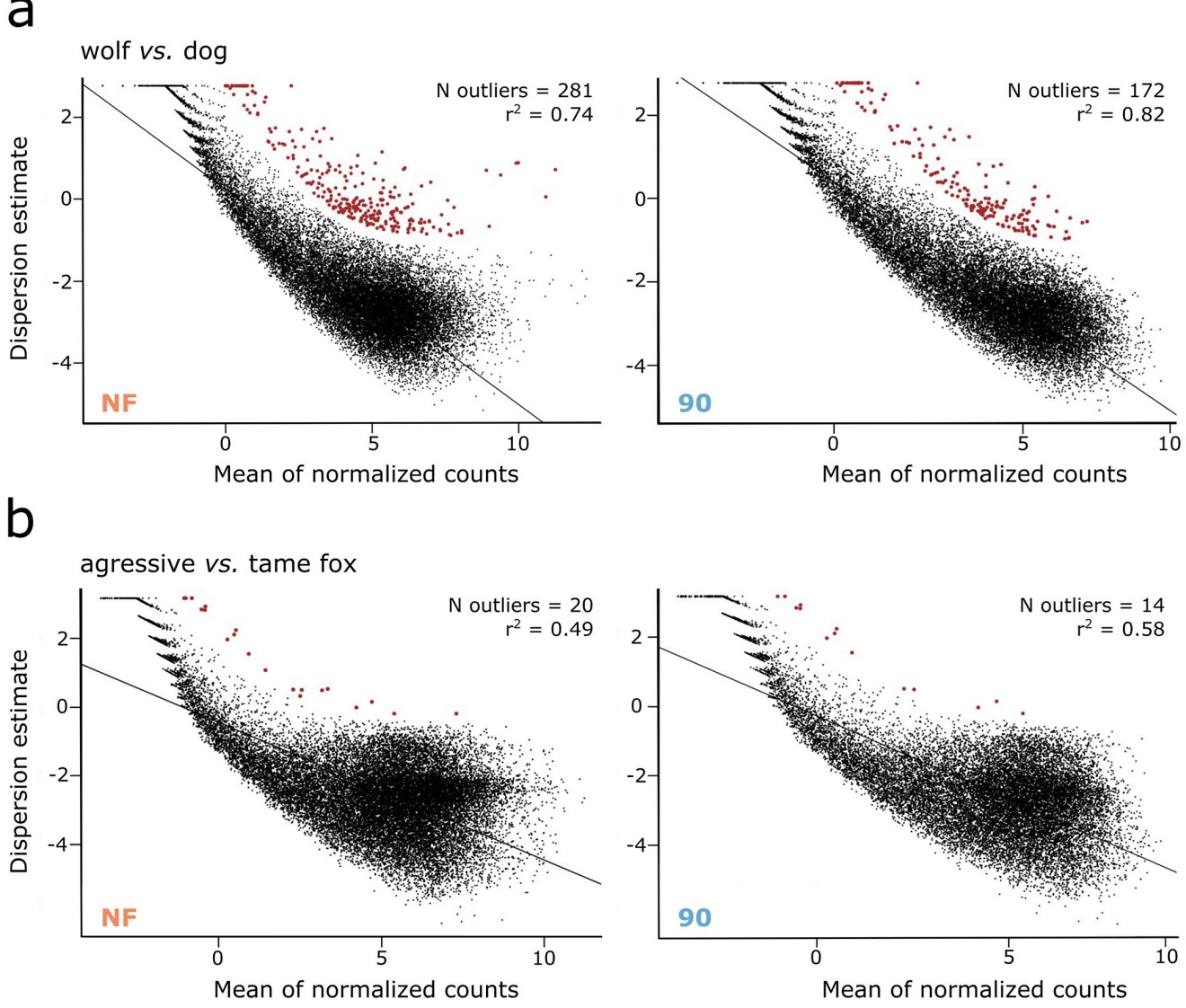

**Fig 6. Distribution of dispersion estimates.** Plots of final dispersion estimates for both case studies, a) wolves and dogs; b) tame and aggressive foxes, calculated using DESeq2 for the non-filtered (NF; orange) and 10% filtered datasets (90th; blue). Each black dot represents a single transcript, and red dots represent outliers. The number of outliers and correlation index ($r^2$) are displayed in the top right corner of each panel. Both x and y-axis are transformed into a logarithm scale. The line in each graph corresponds to the regression analysis between the mean of normalized counts and dispersion estimates.

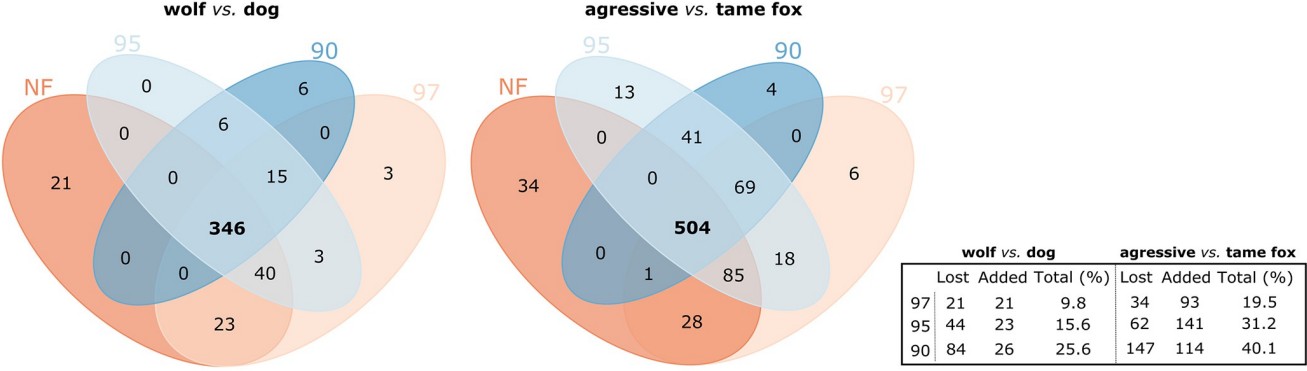

**Fig 7. Differentially expressed transcripts overlapping between non-filtered and filtered datasets.** Venn diagrams representing the number of overlapping differential expressed transcripts found following differential expression analysis using non-filtered datasets and filtered datasets (97th, 95th and 90th percentiles), within each case studies. The inset table provides information about the number of differential expressed transcripts lost/added following each filter step in relation to the non-filtered dataset as well as the the percentage of total discordance.

the highest levels of intra-condition variation a reduction in the number of outliers within both case studies was also observed (Fig 6a and 6b; S9 Table).

**(iii) Discordance lists of differentially expressed transcripts between applied filter levels.** Within the wolves *vs.* dogs case study, from the 430 differentially expressed transcripts identified when using non-filtered data as input to DESeq2, 346 were maintained when using input data filtered at the 97th, 95th, and 90th percentile threshold values (Fig 7 and S7 Table). 26 transcripts were added as differentially expressed following filtering. For this case study, the overall discordance between the differentially expressed transcripts identified using filtered and non-filter input data was 25.58% (Fig 7—inset table). For the second case study, aggressive *vs.* tame foxes, 504 out of the 651 differentially expressed transcripts identified using the non-filtered inputs were maintained when using filtered input data at the 97th, 95th, and 90th percentile threshold values, with up to 114 being added following filtering (Fig 7 and S8 Table). This time the overall level of discordance was 40.09% (Fig 7—inset table). Importantly, in both case studies the added transcripts were consistently maintained across the three filter levels. This reflects the general tendency observed within iterative testing using simulated data where the differentially expressed transcripts identified using lower filter levels are maintained at higher levels of filtering in addition to any newly identified transcripts (S4 and S5 Tables).

## Candidate genes and gene families

By performing annotation using the filtered datasets where the number of differentially expressed transcripts was maximized (3% and 5%, in dogs and tame foxes, respectively), we found 21 gene families in common among the up-regulated genes in dogs and tame foxes. These 21 gene families contained 50 genes (Table 1), of which 19 were exclusive to dogs while 24 were exclusive to tame foxes. The remaining seven genes (RGR, CHRNA5, SQLE, ARHGAP25, ITGA7, MYO7A and TRIB2), were found to be commonly up regulated in both dogs and tame foxes. When batch effects based on tissue were considered RGR, CHRNA5, MYO7A and TRIB2 were maintained as being commonly up regulated (S6 Fig). Note: although in relation to the latter SQLE, ARHGAP25 and ITGA7 were lost, batch effects based on tissue across multiple studies where little other batch information is obtainable could be considered as a very conservative exclusion.

In addition, we also found three gene families, containing four genes, simultaneously down regulated in both groups (Table 2). Two of these genes (STMND1 and OASL) were shared

**Table 1. Shared genes and gene families (Up regulation).** List of the gene families, and shared genes, that were commonly up regulated in dogs and tame foxes. The number, and name, of the genes within each gene family are provided, with the corresponding log2fold-change values in brackets for each species. Within each family, single genes were characterized as shared between dogs and tame foxes (bold), or as exclusively to each of the two groups. When more than one transcript for a specific gene was present, all the log2FC values are reported.

| Gene Family | Group | N of genes | Gene name and log2FC value |
|---|---|---|---|
| Retinal G protein-coupled receptor | **Shared** | 1 | **RGR** (2.10 in dogs, 0.78 in tame foxes) |
| Cholinergic receptor nicotinic alpha | **Shared** | 1 | **CHRNA5** (1.1 in dogs, 0.4 in tame foxes) |
| Squalene epoxidase | **Shared** | 1 | **SQLE** (0.54 in dogs, 0.31 in tame foxes) |
| Rho GTPase activating protein | **Shared** | 1 | **ARHGAP25** (0.86 in dogs, 0.72 in tame foxes) |
| | Tame fox | 2 | ARHGAP4 (0.64); ARHGAP30 (0.57) |
| Integrin alpha subunits | Dog | 3 | ITGA6 (1.25, 1.24); ITGA8 (1.14, 0.90); ITGAX (0.97) |
| | Tame fox | 1 | ITGAL (0.73) |
| | **Shared** | 1 | **ITGA7** (0.76 in dogs, 0.46 and 0.49 in tame foxes) |
| Myosin | Dog | 1 | MYO3A (1.12) |
| | Tame fox | 3 | MYOZ1 (1.53); MYO1F (0.93); MYO1C (0.47) |
| | **Shared** | 1 | **MYO7A** (0.82 in dogs; 0.41 in tame foxes) |
| Tribbles pseudokinase | Tame fox | 2 | TRIB1 (0.94); TRIB3 (0.78) |
| | **Shared** | 1 | **TRIB2** (0.61 in dogs; 0.2 in tame foxes) |
| EF hand calcium binding | Dog | 1 | EFCAB1 (2.59) |
| | Tame fox | 1 | EFCAB2 (0.46) |
| Transcription factor | Dog | 1 | TCF23 (2.04) |
| | Tame fox | 1 | TCF19 (0.63) |
| Adhesion G protein-coupled receptors | Dog | 1 | ADGRG6 (1.45) |
| | Tame fox | 1 | ADGRG1 (0.57) |
| Patatin Like Phospholipase Domain | Dog | 1 | PNPLA4 (1.41) |
| | Tame fox | 1 | PNPLA7 (0.59) |
| SRY-box | Dog | 1 | SOX6 (1.26) |
| | Tame fox | 2 | SOX17(0.84); SOX10 (0.66) |
| Hyaluronan and proteoglycan link protein | Dog | 1 | HAPLN1 (1.15) |
| | Tame fox | 1 | HAPLN3 (0.70) |
| Serine/threonine kinase | Dog | 2 | STK17A (1.15, 1.14); STK32A (1.10) |
| | Tame fox | 1 | STK40 (0.57) |
| Potassium channels | Dog | 1 | KCTD16 (0.98) |
| | Tame fox | 1 | KCTD15 (0.72) |
| Podocalyxin like | Dog | 1 | PODXL (0.95, 0.84) |
| | Tame fox | 1 | PODXL2 (0.70, 0.69, 0.67) |
| ATP binding cassette subfamily B | Dog | 1 | ABCB1 (0.93) |
| | Tame fox | 1 | ABCB9 (0.52) |
| Zinc finger DHHC-type | Dog | 1 | ZDHHC15 (0.75) |
| | Tame fox | 1 | ZDHHC1 (0.70) |
| Sushi domain | Dog | 1 | SUSD1 (0.68) |
| | Tame fox | 2 | SUSD3 (0.79); SUSD6 (0.47) |
| TBC1 domain family | Dog | 1 | TBC1D5 (0.54) |
| | Tame fox | 1 | TBC1D7 (0.27) |
| Mitogen-activated protein kinase kinase kinases | Dog | 1 | MAP3K5 (0.51) |
| | Tame fox | 1 | MAP3K11 (0.76) |

**Table 2. Shared genes and gene families (Down regulation).** List of the gene families, and shared genes, that were commonly down regulated in dogs and tame foxes. The number, and name, of the genes within each gene family are provided, with the corresponding log2fold-change values in brackets for each species. Within each family, single genes were characterized as shared between dogs and tame foxes, or as exclusively to each of the two groups. When more than one transcript for a specific gene was present, all the log2FC values are reported.

| Gene Family | Group | Number of UE | Gene name and log2FC value |
|---|---|---|---|
| Stathmin domain | **Shared** | 1 | **STMND1** (-1.18 in dogs, -0.53 in tame foxes) |
| Oligoadenylate synthetase like | **Shared** | 1 | **OASL** (-0.41 in dogs, -0.52 in tame foxes) |
| Heat shock protein family B | Dog | 1 | HSPB8 (-0.70) |
| | Tame fox | 1 | HSPB11 (-0.32) |

between dogs and tame foxes, while the other two were unique to each group. When batch effects based on tissue were taken into account, STMND1 and OASL were maintained as being commonly down regulated (S6 Fig). The same analysis performed using the non-filtered datasets revealed similar results (S10 Table), although the RGR gene family which included a shared gene between dogs and tame foxes, was lost. This gene was not differentially expressed in the non-filtered fox dataset, representing an example of genes added as differentially expressed after filtering.

## Discussion

Studies involving RNA-seq data often rely on the identification of one, few or many differentially expressed transcripts in order to draw conclusions about biological pathways or about general transcriptome function and evolution. The explicit quantification of intra-condition count variation associated with such transcripts is important for maintaining the context of ambiguity that may exist following differential expression analysis. This is especially true given the growing ability to base highly informative studies around archived transcriptomics datasets at an inter-study level. Here, we developed a method that quantifies intra-condition variation for each individual transcript within the reference set and that can be used to explore the effects of identifying and removing reference-based transcripts harbouring such variation above specified thresholds. By initially applying the method to extensive highly controlled simulated datasets harbouring pre-defined levels of intra-condition count variation we demonstrate the high effectiveness of DESeq2 in identifying differentially expressed transcripts, but also that it can be advantageous to reduce intra-condition variation within the count datasets in relation to identifying additional differentially expressed transcripts that could have been overlooked without such filtering (Fig 2 and S5 Fig, S4 and S5 Tables). By using highly controlled simulated datasets for initial testing, we also provide a tentative estimate on the limit of random intra-condition count variation above which the ability to reliably detect differentially expressed transcripts is diminished (Fig 2 and S5 Fig).

Our real data case study showed that, on average, nearly 20% of the transcripts identified as being differentially expressed prior to filtering contained levels of intra-condition variation equal to or above the 90[th] percentile value of the total distribution. This was higher than the relative proportion of such transcripts within the reference set and indicates that transcripts associated with higher intra-condition variation have a tendency to being identified as differentially expressed. When transcripts possess large amounts of such variation, some ambiguity in their identification as being differentially expressed is inevitable, since reliable expression patterns for at least one of the two conditions being compared have not been fully established; even if statistical correction is applied. This likely partially explains the level of discordance between various differential expression tools available [15, 23, 26–29], for which no consensus

on the best approach to apply exists [30]. However, more importantly, when such transcripts are used for drawing biological conclusion, the context of this uncertainty must be maintained.

We then explored the effects of removing transcripts associated with intra-condition variation, at varying threshold levels, on the gene-wise dispersion estimates, used by DESeq2. Within both case studies, as such transcripts were increasingly removed from input datasets prior to differential expression analysis, the correlation between the mean of normalized counts and dispersion estimates increased, and the number of outliers identified decreased (Fig 6a and 6b; S9 Table; S7 Fig). This, along with discordances between the lists of differentially expressed transcripts identified prior to and post filtering, suggests that transcripts were not simply removed because of physical exclusion from the input data, but that they were also removed, and added, as a result of the effects of removing intra-condition variation from the general gene-wise dispersions applied. The high rates of discordance we found, reaching 40% within the aggressive *vs.* tame fox case study (Fig 7 and S8 Table), reveal how dependent the identification of differentially expressed transcripts is on the accuracy of gene-wise dispersion estimates used; these in turn being affected by transcripts associated with high intra-condition count variation.

High intra-condition count variation at an inter, and to a lesser extent intra, study level can arise from a range of sources including i) biological differences between samples such as age, sex, diet, and health; ii) *in silica* error involving assembly tools producing poorly understood chimeras within the reference transcriptome [50, 60, 61]; iii) ambiguities in read mapping to such references [62]; iv) normalization of count data derived from such mapped reads [63]; and v) including *in vitro* error during library preparation protocols [64, 65]. Although we used DESeq2 within our study, the results of our exploration on the effects of intra-condition variation in the detection of differentially expressed transcripts likely applies to other software used for differential expression analysis that rely on per transcript count information across replicates for the estimation of transcript abundance and dispersion, for example, edgeR [10], BBSeq [66], DSS [67], baySeq [68] and ShrinkBayes [69].

Following the removal of the 3% and 5% of transcripts associated with the highest levels of variation between wolves and dogs, and aggressive and tame foxes, respectively, we observed an increase in the number of differentially expressed transcripts. This pattern is similar to what we observed within our extensive iterative differential expression analysis experiments on simulated data where the levels of intra-condition variation, as well as sets of count over represented transcripts, were explicitly controlled. Thus, this result suggests that for our case studies the removal of variation at these levels optimized the detection of differentially expressed transcripts whilst maintaining consistency. Using these 3% and 5% cut-offs, amongst the 50 over expressed genes identified, across the 21 shared gene families, seven genes were shared between dogs and tame foxes (Table 1). Of these seven genes, three main functions related to brain development, neurotransmission, and immune response were identified. These functions have been repeatedly associated with behavior selection during domestication by different approaches, such as QTL analysis [40, 70, 71], whole-genome sequencing [72–74], and RNA data both using microarrays and RNA-seq [36, 37, 75–77].

Up until recently, almost no gene overlap had been observed between gene expression profiles involving pairs of domesticated and wild animals [35]. However, a recently published paper performing population genomic and brain transcriptional comparisons in seven bird and mammal domesticated species has revealed a strong convergent pattern in genes implicated in neurotransmission and neuroplasticity [42]. These functions are compatible with those found in our analysis. The shared gene ITGA7 belongs to a gene family that is known to

play an essential role in the control of neuronal connectivity [78] and the inflammatory response [79]. Other genes from this family, for example, ITGA8, have been previously observed to be over expressed in tame foxes [76], and here we also observed its over expression in dogs providing further evidence of the family's role in tameness. Similar functions are associated with the shared genes CHRNA5 [80, 81] and TRIB2 [82] from the cholinergic and tribbles family, respectively. Additionally, we found a shared gene involved in sensing local environmental stimuli, the MYO7A, whose mutation results in loss of hearing and vision [83]. Amongst the three gene families identified as under expressed (Table 2), we found the shared gene STMND1, which deficiency in the amygdala of mice was connected to a deficiency in innate and learned fear [84], a behavior that speculatively could also have an important role in domestication. Although we are aware that this overlap analysis between genes that show the same direction of expression in both dogs and tame foxes is not a formal test for gene convergence, we identified genes involved in several functions previously validated in the scope of domestication.

In this work, we have presented TVscript, a new tool that identifies and removes transcripts associated with high levels of intra-condition variation from RNA-seq count data prior to differential expression analysis. By applying TVscript to simulated data, as well as to real data derived from brain samples of dogs, wolves, tame and aggressive foxes, we demonstrate that as hyper variable transcripts are removed the ability to detect differentially expressed transcripts increases in a robust and repeatable manner. Furthermore, we show that above a certain level of random intra-condition count variation, the identification of differentially expressed transcripts is no longer viable. We propose that studies using RNA-seq data at an inter, or intra, study level should determine whether or not transcripts identified as being differentially expressed, using pre-filtered reference sets, are still identified once filtering based on intra-condition count variation as been performed; regardless of the differential expression software used (or the method of obtaining initial counts). Discussion of such transcripts can then be presented relative to the context of such filtering, thus taking a step forward in reducing the ambiguity surrounding intra-condition count variation. Such context is likely going to be dataset specific, as indicated between differences between our case studies, as the extent of intra-condition count variation will differ between datasets and will rarely be known as a prior to analysis. The latter is further highlighted by the consistent patterns observed during the iterative simulations that we performed where levels of intra-condition variation were pre-specified. Finally, by comparing the genes that were differentially expressed in the brain of dogs and tame foxes, we provided further tentative support for candidate genes involved with several functions long known for being involved with domestication. These genes, and functions, have potential for being involved with selection for tameness, which appears to have played a crucial role in canine domestication. We use the word tentative to describe our support, as the primary aim of this study was to investigate the effects of intra-condition count variation on the detection of differentially expressed transcripts, and the identification of genes involved within an evolutionary process, such as domestication, should be supported by datasets specifically generated for that purpose, and confirmed relative to the different reference transcriptomes involved. The quality of such transcriptomes in turn, in relation to chimeras, missing transcripts and partial redundancies, must also be carefully explored.

## Supporting information

**S1 Fig. Alignment rates obtained using Bowtie2.** Mapping success rates (%) resulting from the alignment of the 44 samples used in this study to the complete dog transcriptome. For each sample, the percentage of aligned reads is presented by the blue bars, while the percentage of

reads failing to map is represented in red (the number of raw reads is available in S1 Table).
(TIF)

**S2 Fig. Correlation between per-transcript counts obtained following Bowtie2 mapping and count estimates obtained using kallisto.** $R^2$ values describing the linear correlation between each count dataset produced from the mapped datasets presented in S1 Fig and corresponding count extimates produced when pseudo-mapping the same RNA-Seq data to the complete dog transcriptome using kallisto.
(TIF)

**S3 Fig. Re-mapping fox data to the fox reference transcriptome.** Read mapping rates achieved when mapping the fox RNA-Seq datasets to the fox reference transcriptome.
(TIF)

**S4 Fig. Transcripts identified by DESeq2 as being over expressed in the absence of randomly introduced intra-condition variation.** Across one hundred iterations the dots represent the number of transcripts identified as being over expressed between condition A and B. Each condition contained five replicates. (A) The one hundred transcripts selected for read over representation within replicates of condition B were maintained as constant and (B) the one hundred transcripts selected for read over representation within replicates of condition B were re-selected during each iteration. During each iteration the ten count datasets that were simulated each reflected even transcript coverage of 3 million read pairs with the exception of the one hundred transcripts selected for over representation in condition B whose count values were increase by a factor of two.
(TIF)

**S5 Fig. Over expressed transcripts pre- and post-filtering (transcripts selected for count over representation were re-selected during each iteration).** The number of transcripts identified by DESeq2 as being over expressed both prior to (light gray) and post (dark gray) filtering within each of the one hundred iterations performed at each level of introduced random intra-condition count variation. Each iteration involved initially simulating ten count datasets divided into conditions A and B following which DESeq2 was run to attempt to identify the one hundred transcripts selected for over representation as described in the methods. Following this the ten simulated datasets were filtered using TVScript with a 95th percentile threshold in order to generate corresponding filtered datasets (divided into corresponding conditions A' and B') on which DESeq2 was re-run.
(TIF)

**S6 Fig. Confirmation of shared genes within differential expression analysis taking tissue effects into account.** The upper dark grey circle contains the nine genes identified as being either commonly over, or under, expressed simultaniously within dogs and tame foxes using filter levels the 95th and 97th percentiles whilst only accounting for condition (wolves *vs.* dogs and aggressive *vs.* tame fox). Six of these genes (RGR, CHRNA5, MYO7A, TRIB2, STMND1 and OASL) are present when DESeq2 is run whilst also accounting for differences in tissue (light grey left oval). SQLE, ARHGAP25 and ITGA7 are observed only within the differentially expressed transcript list that is based solely on condition (dark grey right oval).
(TIF)

**S7 Fig. Distribution of dispersion estimates.** Plots of dispersion estimates in relation to the mean of normalized counts for both case studies, wolves and dogs (left panels), and tame and aggressive foxes (right panels). Estimates were calculated using DESeq2 for the non-filtered (NF) and all filtered datasets (99th, 95th and 90th are shown as an example). Gray dots represent

the gene-wise maximum likelihood estimates (MLE), the red curve shows the fit to the MLEs, and blue dots identify the final maximum *a posteriori* (MAP) estimates of dispersion. Red dots represent the outliers detected by DESeq2. Both x and y-axis are transformed into a logarithm scale.
(TIF)

**S1 Table. Dataset description.** Full details of all datasets, including the location of the relative tissue, age, and sex of each individual, replicate information and sequencing details (FC–frontal cortex; CC–cerebral cortex; PFC–prefrontal cortex; FL–frontal lobe; NS–not specified; F–female; M–male; AD–adult; ya–years old; PE–paired-end; SE–single end).
(DOCX)

**S2 Table. Common over expressed transcripts pre- and post-filtering (when transcripts selected for count over representation are kept constant).** The number of transcripts from the dog reference set that are commonly identified by DESeq2 as being over expressed within condition B both prior to and post filtering for each of the one hundred iterations performed at each level of introduced random intra-condition count variation. Each iteration involved simulating ten count datasets divided into conditions A and B following which DESeq2 was run to attempt to identify the one hundred transcripts selected for over representation as described in the methods section. Filtering involved running TVScript with a 95th percentile threshold on the non-filtered datasets to generate corresponding filtered datasets (divided into corresponding conditions A' and B') following which DESeq2 was re-run and the results compared back to those obtained for the non filtered data.
(DOCX)

**S3 Table. Common over expressed transcripts pre- and post-filtering (when transcripts selected for count over representation are re-selected during each iteration).** Same as S2 Table but where the one hundred transcripts selected for over representation within condition B are re-selected during each iteration.
(DOCX)

**S4 Table. Ratio between the common number of over expressed transcripts pre- and post-filtering and the maximum number detected when transcripts selected for count over representation are kept constant.** Numbers in S2 Table were divided by the maximum number of over expressed transcripts detected within each correspomding iteration i.e. the maximum number detected using corresponding non-filtered and filtered datasets.
(DOCX)

**S5 Table. Ratio between the common number of over expressed transcripts pre- and post-filtering and the maximum number detected when transcripts selected for count over representation are re-selected during each iteration.** Numbers in S3 Table were divided by the maximum number of over expressed transcripts detected within each correspomding iteration i.e. the maximum number detected using corresponding non-filtered and filtered datasets.
(DOCX)

**S6 Table. Removal of intra-condition variation.** Number of transcripts kept and removed from the reference in each case study, wolves and dogs, and aggressive and tame foxes, across the filtered levels used (from the 99th to the 70th percentile). The first ten percentiles were explored in greater detail in steps of one, while the remaining were performed in steps of 5.
(DOCX)

**S7 Table. Differentially expressed transcripts in wolf *vs*. dog.** Complete list of differentially expressed transcripts in dogs when compared to wolves, identified using non-filtered datasets, and those that got removed (red) within the highest 10% of intra-condition variation, as well as those added (green) as differentially expressed across selected filtered datasets (97th, 95th, and 90th percentiles). The correspondent annotated gene ID, log2FC values and p-values are provided.
(DOCX)

**S8 Table. Differentially expressed transcripts in aggressive *vs*. tame fox.** Complete list of differentially expressed transcripts in tame foxes when compared to aggressive foxes, identified using non-filtered datasets, and those that got removed (red) within the highest 10% of intra-condition variation, as also those added (green) as differentially expressed across selected filtered datasets (97th, 95th, and 90th percentiles). The correspondent annotated gene ID, log2FC values and p-values are provided.
(DOCX)

**S9 Table. Correlation and outliers.** Correlation values ($r^2$) and the root mean square error (RMSE) from the regression analysis between the final dispersion estimates and the mean of normalized counts for both case studies, wolves and dogs, and aggressive and tame foxes. The number of outliers identified by DESeq2 are also presented. Values are shown for the non-filtered (NF) and all the filtered datasets used in differential expression analysis.
(DOCX)

**S10 Table. Shared genes and gene families between non-filtered datasets.** List of the gene families, and shared genes, that were commonly regulated in dogs and tame foxes, using the non-filtered datasets. The number, and name, of the genes within each gene family are provided, with the corresponding log2fold-change values in brackets for each species. Within each family, single genes were charecterized as shared between dogs and tame foxes, or as exclusive to each of the two groups. When more than one transcript for a specific gene was present, all the log2FC values are reported.
(DOCX)

## Author Contributions

**Conceptualization:** Diana Lobo, Raquel Godinho, John Patrick Archer.

**Data curation:** Diana Lobo, Raquel Godinho.

**Formal analysis:** Diana Lobo, Raquel Linheiro, John Patrick Archer.

**Funding acquisition:** Raquel Godinho, John Patrick Archer.

**Investigation:** Diana Lobo, Raquel Godinho, John Patrick Archer.

**Methodology:** Diana Lobo, John Patrick Archer.

**Project administration:** John Patrick Archer.

**Resources:** Raquel Godinho.

**Software:** Diana Lobo, John Patrick Archer.

**Supervision:** Raquel Godinho, John Patrick Archer.

**Validation:** Diana Lobo, Raquel Linheiro, John Patrick Archer.

**Visualization:** Diana Lobo, Raquel Linheiro.

**Writing – original draft:** Diana Lobo, Raquel Godinho, John Patrick Archer.

**Writing – review & editing:** Diana Lobo, Raquel Linheiro, Raquel Godinho, John Patrick Archer.

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
