## [Decision Letter · Decision Letter 0]

23 May 2022

PONE-D-21-32424

On taming the effect of transcript level intra-condition count variation during differential expression analysis: a story of dogs, foxes and wolves

PLOS ONE

Dear Dr. Archer,

Thank you for submitting your manuscript to PLOS ONE. After careful consideration, we feel that it has merit but does not fully meet PLOS ONE’s publication criteria as it currently stands. Therefore, we invite you to submit a revised version of the manuscript that addresses the points raised during the review process.   You will see that the reviewers have raised some concerns regarding the methodology, specifically the choice of data for evaluation and the possibility of batch effects in these data, which will need to be addressed. In addition several aspects of the methodology require further clarification.

 Please note that PLOS publication criteria only require a study to be rigorous, robust and described in sufficient detail for replication. Therefore, while I agree with reviewer 1 that an R package may improve user uptake of your tool, this is not required for acceptance of your manuscript, since both reviewers have confirmed you have made usable code available.

We look forward to receiving your revised manuscript.

Kind regards,

Katherine James, Ph.D.

Academic Editor

PLOS ONE

Journal Requirements:

3. Please upload a new copy of Figure S1 and S2 as the detail is not clear. Please follow the link for more information: https://blogs.plos.org/plos/2019/06/looking-good-tips-for-creating-your-plos-figures-graphics/" https://blogs.plos.org/plos/2019/06/looking-good-tips-for-creating-your-plos-figures-graphics/

Reviewers' comments:

Reviewer's Responses to Questions

**Comments to the Author**

1. Is the manuscript technically sound, and do the data support the conclusions?

Reviewer #1: Partly

Reviewer #2: Partly

2. Has the statistical analysis been performed appropriately and rigorously? 

Reviewer #1: Yes

Reviewer #2: No

3. Have the authors made all data underlying the findings in their manuscript fully available?

Reviewer #1: Yes

Reviewer #2: Yes

4. Is the manuscript presented in an intelligible fashion and written in standard English?

Reviewer #1: Yes

Reviewer #2: Yes

5. Review Comments to the Author

Reviewer #1: Authors outline that filtering expression data based on intra-group variation is recommended for maximising the number of identified DE genes. However the goal of DE analysis is not to maximise the number of DEGs but to identify the truly correct DEGs, those that are likely to be replicated if the experiment were conducted again or confirmed with another technique. In addition, genes that have high variability may actually be true DE genes, and there is no valid reason to discard them. To build a better justification for such filtering, a more comprehensive analysis is required to show that accuracy in DE classification is improved. Analysis of RNA-seq datasets with large numbers of replicates would be useful (eg: PMID: 27022035).

I downloaded the package and tried the example. It seemed to work fine using the directions in the manual.

The genes which are hypervariable in expression, are these markers of different brain regions? I ask because dissection and sampling can be a major source of variation.

P9: Regarding the way the variance is calculated, is it calculated for each sample group separately and then the average of the two groups is used, or is this done in a different way?

Typically, in order to avoid violating FDR correction assumptions, it is not allowed to filter any genes after the sample labels have been revealed as this equates to cherry picking, a form of p-hacking. In microarray analysis it is customary to discard probes with low overall variance but is acceptable as this procedure does not peek at the sample labels before filtering (eg: PMID: 19133141). Some analysts filter lowly expressed RNA-seq data using a threshold of 1 TPM or an average of 10 reads per sample on average which is also fine.

DESeq2 and other differential expression tools are written in R so it makes sense that this tool would also be written in R. Exporting the R data objects as TSV, running tvscript and then reading the data back into R is clumsy and may lead to poor uptake of this tool. I’d recommend a bioconductor package, which has the added benefit of being able to generate charts so that the user can better understand the intra-condition variability, like how edgeR generates a BCV chart (https://rdrr.io/bioc/edgeR/man/plotBCV.html). Another informative diagnostic chart could be PCA plots of (1) all transcripts, (2) hypervariable discarded transcripts, and (3) retained transcripts.

Bowtie not recommended for transcriptome mapping. As there are reads that can map equally well to multiple transcripts which get discarded in such approaches, it is preferable to use Kallisto or Salmon which deals accurately with multi-mapped reads. This may explain the reason behind the low mapping rate of wolf, dog and fox reads.

Does this approach work equally well for gene-based analysis using counts generated using STAR or featureCounts?

This does not sound right. “The number of transcripts removed was higher for the fox samples than for those of wolf and dog, reflecting the higher intra-condition variability present.” If percentiles of transcripts are being discarded, shouldn’t the proportion of detected transcripts discarded be the same for both studies? It is not explained clearly.

The figures should be explained in sequence. Eg: Figure 4C and the minitable in Fig 4 should be explained in the text before Fig 5a.

Reviewer #2: Lobo et al present an evaluation of their software TVscript, which evaluates intra-condition variability in the counts that have been mapped to a transcriptome in an RNA-Seq experiment and removes the transcripts associated with the highest level of this variability, up to a user specified percentile threshold. They test the software by applying it to two pairs of datasets from wild and tame animals, wolves vs dogs and aggressive vs tame foxes. The greatest fraction of differentially expressed transcripts (DETs) is obtained by removing 3 to 5% of transcripts, and the authors describe some interesting features of the gene families of the corresponding differentially expressed genes, including common changes upon taming.

The approach to RNA-Seq analysis is a potentially interesting one, representing another approach similar in some ways to the “orthogonal filtering” of low-expressed transcripts that is commonly used to increase the power in the analysis of RNA-Seq experiments.

Unfortunately there are a number of aspects to the methodology that make it hard to recommend publication in the present form.

1 Most importantly, I don’t think the data sets analysed are appropriate for the main intention of the paper. It is hard to tell whether the alterations made to the transcriptome improve the results rather than inducing false positives. The data sets used for testing come from multiple batches, and two organisms, one of which is appreciably divergent from the transcriptome to which it is aligned. In short, there are too many other uncontrolled factors in the analysis done to tell whether the results are reliable. In testing TVscript, it would be better to use an approach like that taken in Rapaport et al 2013 (https://doi.org/10.1186/gb-2013-14-9-r95), which uses data sets where batch effects are better controlled, including one (GEO GSE 49712) where external rna control consortium (ERCC) spike ins were used to produce known true positive DEGs.

2 More detail is necessary about how the differential expression was performed; figure 2c seems to show that the dogs do separate by batch (1-5; 6; 7; 8-9) and one would normally use a design formula that took account of the different sources of the data, something like ~ batch + tameness.

(although the brain tissue and instrument used are also similar for dogs 6-9, so one could also try ~ tissue + tameness). The authors should state whether they used a formula like this, and justify why not if they did not (Ideally, the R script used for differential expression analysis could be made available).

I would also remark that, since the authors emphasise that data comes from different sources, it was not immediately clear, until I looked at supplementary table 1, that the wolves and dogs 1-5 all come from one study, and similarly all the foxes, aggressive and tame, come from one study. This should be brought out more in the text, as otherwise the reader is made to wonder how any difference between wild and tame will be detectable that is not confounded with batch effects.

3 It is unclear to me why the authors used the C. familiaris transcriptome for their work on the fox as well as the dog/wolf, when the fox and wolf lineages diverged 10 myr ago. A genome and transcriptome are available for the fox (https://www.ensembl.org/Vulpes_vulpes/Info/Index?db=core), and even though it is of lower quality than the dog, a higher mapping rate might have been expected. I appreciate that it makes the assessment of TVscript, and to some extent the comparison of dog and fox DET results, more straightforward (though information on orthology is also available). The low mapping rate and slightly strange clustering of the points in the PCA plot fig 2d are indications there may be some problems with the fox data that might in part come from the choice of transcriptome, and this casts some doubt on the DET results for me.

My recommendation would be to split the work into two papers, one comparing the wild and tame animals, which to me was the most interesting part of the manuscript, and one assessing TVscript. It appears to me from comparing Tables 1 and 2 in the paper (filtered transcriptome) with supplementary table 6 (unfiltered transcriptome), that the filtering did not make a very big difference here. Hence the first paper could use the results from the more standard methods of supp. Table 6 and the interesting overlaps between the changes on domestication in the two pairs of animals would still be largely maintained. The second paper really would really need to use different test data sets , as suggested above, to establish whether TVscript is genuinely increasing sensitivity without introducing type I errors.

I would like to thank the authors for preparing the manuscript carefully, providing detailed results and supplementary material, and providing access to the code of TVScript along with links to other useful material on sourceforge.

6. PLOS authors have the option to publish the peer review history of their article (what does this mean?). If published, this will include your full peer review and any attached files.

Reviewer #1: No

Reviewer #2: No

---

## [Author Response · Author response to Decision Letter 0]

14 Jul 2022

Cover letter to editor and the responses to reviewers are pasted below as requested. These have also been uploaded as separated documents.

Cover letter (also uploaded):

Dear Dr. James,

Ref: Revision of manuscript titled “On taming the effect of transcript level intra-condition count variation during differential expression analysis: a story of dogs, foxes and wolves”. 

We were very pleased to receive the feedback from both reviewers, and from you, in relation to our manuscript. We found the comments valuable in clarifying the content and presentation of our work. We have now responded to each of the comments in turn and have uploaded our modified manuscript, and requested files, as follows:

1. The clean manuscript file titled “manuscript.docx”. This is the version to which the line numbers within the response to reviewers document are relevant.

2. The corresponding file with track changes turned on. This is titled “'revised_manuscript_with_track_changes.docx”.

3. The reviewer response file titled “response_to_reviewers.docx”.

4. This cover letter titled “cover_letter_2.docx”.

As requested, we have invested extensive effort into evaluating our approach. In brief, prior to presenting our results in relation to real data, we now perform a series of differential expression analysis experiments involving highly controlled simulated datasets within which the exact level of background intra-condition count variation could be specified, as well as the specification of a subset-set of transcripts to be over-expressed across replicates within second conditions used. This testing framework was used to detect the known over expressed transcripts relative to incrementing levels of background intra-condition count variation. For each level of variation introduced one hundred iterations of differential expression analysis were performed. Details of this are found in the first response to each of the two reviewers. During testing our approach was shown to have consistent appositive effect on the detection of differentially expressed transcripts. We have added an author Raquel Linheiro, who helped to perform the simulated differential expression experiments, and who also tested for batch effects, within our dog/wolf data. I hope that the addition of an author at this point is acceptable.

In relation to batch effects, we have tested for this, based on tissue as reviewer #2 suggested, and provide the result in the new figure S6. We have also clearly explained within the manuscript why we chose to use comparisons based on condition alone for our dog vs. wolf data (reviewer 2: comment 2). For fox data all samples came from the same batch (and tissue), and for simulations batch effects were not relevant due to the nature in how the datasets were generated.

We have discussed our reasoning for implementing our tool in Java within the manuscript, and appreciate the editors choice in being lenient in relation to this given we have made source code available. We have highlighted within the manuscript that the approach could be readily implemented in R and, if future user demand is there, we are happy to create a supported R version.

In relation to our data availability statement, all data used is publically available on NCBI and is described in detail within table S1, but we have now added the required statement that includes all run accession numbers during the resubmission process. Additionally, all count data from the raw data that we use has been made publically available on the Zenodo platform. All old figures have been checked and re-uploaded along with the new ones.

Once again we found that all suggestions made by the reviewers were very helpful in improving our manuscript, for example, remapping to the fox transcriptome as well as remapping all of our data using kallisto to validate our counts obtained using Bowtie2, and we no hope that they, as well as you, find that our manuscript is now ready for presentation to the PLOS ONE readership. We await (tentatively) a positive response.

Finally in relation to our financial statement, we would like it to read as below, but there is no place on the re-submission platform to input this. The statement associated with the first submission is present on the final PDF build, but we need it replaced by this one: 

“This work was funded by the project NORTE-01-0246-FEDER-000063, supported by Norte Portugal Regional Operational Programme (NORTE2020), under the PORTUGAL 2020 Partnership Agreement, through the European Regional Development Fund (ERDF), and by research funding from the projects under the references PTDC/BIA-EVF/29115/2017, PTDC/BIA-EVF/2460/2014 and POCI-01-0145-FEDER-029115 co-funded by Operational Competitiveness and Internationalization Program, Portugal 2020 and the European Union via the European Regional Development Fund (ERDF) and by National Funds through FCT. DL, RG were supported by FCT (PD/BD/132403/2017 to DL, contract under DL57/2016 to RG) and JA was supported by Funds through FCT under the references POCI-01-0145-FEDER-029115 and PTDC/BIA-EVL/29115/2017. The funders had no role in study design, data collection and analysis, decision to publish, or preparation of the manuscript. FCT and NORTH2020 url’s: https://www.fct.pt/ and https://www.norte2020.pt.”

As before we thank you for your consideration.

Best Regards,

John Archer,

Principal Researcher (Bioinformatics)

CIBIO-InBIO, Universidade do Porto, Campus de Vairão, Rua Padre Armando Quintas. 4485-661 Vairão, Email: john.archer@cibio.up.pt

Response to reviewers (also uploaded):

Reviewer #1: 

1. Authors outline that filtering expression data based on intra-group variation is recommended for maximising the number of identified DE genes. However the goal of DE analysis is not to maximise the number of DEGs but to identify the truly correct DEGs, those that are likely to be replicated if the experiment were conducted again or confirmed with another technique. In addition, genes that have high variability may actually be true DE genes, and there is no valid reason to discard them. To build a better justification for such filtering, a more comprehensive analysis is required to show that accuracy in DE classification is improved. Analysis of RNA-seq datasets with large numbers of replicates would be useful (eg: PMID: 27022035).

We thank the reviewer for providing this detailed review of our manuscript and we have responded to each of the points raised below. We feel that each point has greatly improved our study as well as its presentation. In relation to the initial summarization, we agree with what the reviewer has said, mainly that the goal of DE analysis is not to blindly maximize the number of DEG´s but to identify those that are truly correct. We now clarify this in the manuscript (line 176 of the introduction) and we have added an extensive analysis to demonstrate that we are improving the identification of differential expressed transcripts and not just maximizing the numbers in the lists produced. 

In relation to the latter (which also overlaps with the first comment made by reviewer #2) we extensively explore the effects of intra-condition per-transcript read count variation on the identification of differentially expressed transcripts through a series of differential expression analysis experiments involving highly controlled simulated count datasets derived from the available dog reference transcriptome. These are mentioned within the introduction (line 130) and described in detail within the materials & methods section under the new section titled “Controlled intra-condition variation within simulated data” (starting at: line 247). Using simulated count datasets allowed the exact level of background intra-condition count variation to be specified, as well as the specification of a subset-set of transcripts to be over-expressed across replicates within second conditions used (for each experiment performed). Within our simulated replicates, background count variation refers to varying count levels associated with individual transcripts that are not maintained across replicates (of a given condition), thus effectively reflecting intra-condition noise. On the other hand specifying a subset of transcripts to be over represented across replicates of a condition reflects identifiable over expressed transcripts. 

Following a brief introductory set of differential expression experiments on simulated data where levels of intra-condition variation are kept at zero (line 265), a set of simulations where the levels of background variation explored went from 1% to 10% in steps of one were performed. At each increment one hundred iterations of the following were performed (line 282): (i) Ten replicates of count dataset were generated, representing reads evenly spread across 22,580 dog transcriptome transcripts (after normalization by length) ranging in length from 300 to 5000 bp, and allocated into two conditions A and B (five in each). Within B one hundred selected transcripts had counts over represented. (ii) At the percent level of variation associated with the iteration, that percent of transcripts from each of the ten replicates were randomly selected for count over-representation. (iii) DESeq2 was used on the count files within conditions A and B to obtain a list of over expressed transcripts. (iv) TVscript was run using a 95th percentile variance threshold to generate ten corresponding modified count files separated into two conditions (A’ and B’). (v) DESeq2 was again used on these to obtain a list of over-expressed transcripts. (vi) The lists of over-expressed transcripts obtained in (iv) and (v) were cross compared. This was repeated in two different ways: (i) the one hundred transcripts initially flagged for over representation were kept constant throughout all levels of variation and for each of the associated iterations and (ii) during each level of variation and for each iteration a new set of one hundred transcripts were randomly selected. 

The results are presented within two new paragraphs (staring at: line 414) and within the newly added figures (Fig. 2 and Fig. S5), as well as Tables S2, S3, S4 and S5. In brief, the results in the figures indicate that within each iteration of steps (i) to (vi) there are slightly more of the one hundred transcripts selected for read over representation (within condition B’s of each iteration) identified post filtering. Tables S2 and S3 provide the counts of commonly identified differentially expressed transcripts (before and after filtering) for each of the one hundred iterations, at each level of intra-condition variation introduced. However even if the numbers are high they must be taken in the context of the maximum identified from either pre- or post filtered data. This is why tables S4 and S5 were provided as they present the agreeability relative to these maximums, and from levels of randomly introduced intra-condition count variation going from 1 to 4% the agreeability is high (Table S4 - 1 to 4% averages: 0.96, 0.98, 0.94 and 0.81; Table S5 - 1 to 4% averages: 0.96, 0.97, 0.93 and 0.82) (line 447). Finally, within figure 2 and figure S5, at and above the 5% level of intra-condition variation the ability to successfully identify the one hundred transcripts selected for over representation within condition B greatly diminishes within iterations. This could be indicative of a tentative estimate on the limit of at what level of random intra-condition count variation becomes inhibitory within differential expression analysis studies (line 452).

We thank (both) reviewers for providing dataset suggestions with larger numbers of replicates than we initially used in our analysis and pointing out this area that could be improved. We felt that using simulated data for this newly included analysis allowed us to explicitly control all relevant factors involved and allowed us to recreate the many replicates under the varying degrees of intra-condition variation described. We feel that this comment has contributed greatly to the clarity of our exploration in relation to the effects of intra-condition read count variation on the detection of differentially expressed transcripts.

2. I downloaded the package and tried the example. It seemed to work fine using the directions in the manual.

We thank the reviewer for taking the time to test our tool. As an aside point, future work is planned on developing a graphical user interface and increasing in-software visualization and interaction with the input count datasets. We have also contemplated in allowing the user to input raw read datasets and perform tasks such as estimating counts internally using our own implemented pseudo-mapper or those such as kallisto (mentioned by this reviewer later). We have not altered the software since the previous review but we do explicitly state that it is open source at the end of the introduction (line 155).

3. The genes which are hypervariable in expression, are these markers of different brain regions? I ask because dissection and sampling can be a major source of variation.

It is correct that across different brain regions one would expect differing levels of expression and this is the reason why we limited compartments included frontal cortex, cerebral cortex, prefrontal cortex and frontal lobe for dogs and wolves and just prefrontal cortex for aggressive and tame foxes. In addition to this information being provided in Table S1, we have clarified the compartments that we use within the Materials & Methods section (line 162 and 166). We also now explicitly mention the possible effects of differing compartments and batch effects relative to the wolf and dog data (starting at: line 320); also in relation to a comment made by the second reviewer. Finally, as the reviewer points out, dissection and sampling can be one major cause of intra-condition variation, that will subsequently have an impact on the detection of DEGS. This is true even for biological replicates at an intra-study level and the reason why we opted for simulated data during testing, where we had explicit control in relation to specifying which transcripts were to be over represented as well as the levels of background count variation, in relation to the first comment by this reviewer.

4. P9: Regarding the way the variance is calculated, is it calculated for each sample group separately and then the average of the two groups is used, or is this done in a different way?

In the Materials & Methods paragraph, titled Software, it is mentioned that (≈line 210): “ … (iii) for each reference transcript (t), the absolute pairwise differences between normalized read counts across all samples within condition A are calculated; (iv) the corresponding variances are calculated; (v) steps (iii) and (iv) are repeated for condition B; (vi) variance scores from each condition are placed in ascending order and associated with corresponding percentiles; …” Thus, variance scores are calculated for each group separately, following which all scores from each group are placed in a sorted list. It is this final sorted list that is used for calculating thresholds. This way scores from both groups are represented within the final distribution used to calculate percentiles on and there is no averaging involved. We have clarified this within this paragraph (line 226).

5. Typically, in order to avoid violating FDR correction assumptions, it is not allowed to filter any genes after the sample labels have been revealed as this equates to cherry picking, a form of p-hacking. In microarray analysis it is customary to discard probes with low overall variance but is acceptable as this procedure does not peek at the sample labels before filtering (eg: PMID: 19133141). Some analysts filter lowly expressed RNA-seq data using a threshold of 1 TPM or an average of 10 reads per sample on average which is also fine.

We are exploring the effects of intra-condition variation at a per transcript level, simultaneously across all transcripts, on the general ability to detect those that are differentially expressed (line 103). We agree that our initial datasets were not sufficient to confirm, as we had no control on the actual levels of per-transcript intra-condition count variation that were present, nor did we have a list of specified transcripts that were guaranteed to be over expressed. Additionally we had no way to verify across hundreds of repetitions the consistency of the results. This would be largely true for any “sequenced” datasets used. We feel that with the addition of the simulated study, where parameters were explored in a highly defined framework, we are approaching a point where our exploration is more interesting to a wider readership as we could specify the actual level variation present within the replicates (in conditions A and B) of each differential expression experiment performed (iterations) involving both TVScript filtered and non-filtered counts and verify result across hundreds of iterations. 

We do not feel that we are cherry picking transcripts, but instead using a robust variation threshold framework based on percentiles to explore the effects of intra-condition variation. We initially selected the 95th and 97th percentiles for such cut-offs as these maximized the number of differentially expressed transcripts, but admittedly we had not previously demonstrated the robustness of the approach though a simulated study (now included as described above), nor that then additional transcripts were on top of those identified prior to filtering as a result of the reduction of noise within the data. We hope now that with the addition of the extensive simulations involving predefined over expressed transcripts that the reviewer is convinced that there is no cherry picking involved and that the increase in the over expressed transcripts is because of the optimization of the threshold for removing noisy transcripts. 

Finally, we fully admit that this is an exploratory approach and that optimized threshold will vary depending on the datasets involved thus potentially limiting general applicability without a complete prior exploration of the datasets involved in a similar manner to that presented here. We have now explicitly stated this with the concluding paragraph of the discussion (line 723). We feel that exploring and highlighting the effects of intra-condition count variation is still a highly relevant, and interesting, study to make widely available as its context is often mute within differential expression analysis studies, even though the transcript lists produced are at times a major part of the end result. Disagreeability between approaches used to identify differentially expressed transcripts is further evidence of the effects of such variation (line 95).

6. DESeq2 and other differential expression tools are written in R so it makes sense that this tool would also be written in R. Exporting the R data objects as TSV, running tvscript and then reading the data back into R is clumsy and may lead to poor uptake of this tool. I’d recommend a bioconductor package, which has the added benefit of being able to generate charts so that the user can better understand the intra-condition variability, like how edgeR generates a BCV chart (https://rdrr.io/bioc/edgeR/man/plotBCV.html). 

Yes, many differential expression analysis tools are written in R, but prior to using them an array of many non-R based tools are required, for example mapping or abundance estimators (such as Bowtie2 or kallisto). For this reason we thought that having TVscript in a platform independent language such as Java would be acceptable. Additionally, we wanted the potential to be able develop an optional graphical user interface that could sit on top of our tool and help visualize, samples, count data and output files in a rich transcriptome analysis environment that would be difficult to provide in a package such as R. That said, our method is not overly complex and, being clearly described within the manuscript in a step wise fashion, it could be readily implemented within R. Eventually, if our approach is accepted as an interesting way to explore such data and if there is enough demand for an R implementation, we would be pleased to provide one, but for now we hope that the reviewer can accept our Java implementation. In relation to this comment we have added the following to the manuscript (line 242).

“Although TVscript is implemented in Java the steps involved can be readily implemented within any language (e.g. R or python), using the detailed description provided above as well as the Java source code that is fully available. There are no dependent packages where code is unavailable. At the time of development we choose Java mainly due to its platform independence, which can be an advantage within setting up analysis pipelines involving many different tools. That said we are aware that many differential expression analysis tools are R based and future demand may warrant a supported R version.”

7. Bowtie is not recommended for transcriptome mapping. As there are reads that can map equally well to multiple transcripts which get discarded in such approaches, it is preferable to use Kallisto or Salmon which deals accurately with multi-mapped reads. This may explain the reason behind the low mapping rate of wolf, dog and fox reads.

In relation to the placement of reads that map to identical regions of transcripts, we are not under the impression that kallisto deals with these more reliably, as kallisto is effectively estimating abundance counts for each transcript using kmer summarizations in order to rapidly speed up the obtaining of read counts used for downstream analysis (as read locational placements within transcripts are not found). Kallisto in effect somewhat avoids the problem by not placing individual reads. For example, within a given transcript if there are ambiguous regions present for full mapping, then as long as a pseudo-mapping tool is not looking to specifically place a read the count can be incremented for each read located within that transcript - regardless of the actual location. Kallisto is a very nice piece of software, performs exceptionally well in terms of obtaining transcript count abundances, and in relation to this comment we have mapped all of our data again using kallisto to the dog transcriptome reference set used within our analysis (line 196). 

For each dataset mapped we have summarized the correlations between the Bowtie2 counts that we utilize and the kallisto estimated abundances. In all cases these correlations are exceptionally high and we have discussed this within the results section of the manuscript (line 401) and included a new supplementary figure (Fig. S2). All read counts and abundances by Bowtie2 and kallisto have now been made available on the Zenodo repository (line 405) (https://zenodo.org/record/6778429). High correlations were achieved as Bowtie2 also performs well at mapping RNA-seq data to reference transcriptomes and, even within transcripts where there can be ambiguity in mapping minority numbers of reads, the over all count associated with the transcript obtained using the BBMap package is still accurate.

Tangentially, the dog, fox and wolf datasets that we use in our study are mapped to the available dog reference transcriptome where there are no introns present to disrupt mapping process, and this is why we did not use a splice aware mapper such as Tophat2 or HISAT2. We now say this (line 183).

8. Does this approach work equally well for gene-based analysis using counts generated using STAR or featureCounts?

Such an exploration can be performed as long as count data (or estimated count data) is available that can be allocated into two different conditions. We mentioned this within the concluding paragraph (line 720). The important thing is to make sure the counts are reliable and this is why the reviewers previous kallisto comment was relevant, and why we remapped all our data using that tool to confirm that counts obtained following the Bowtie2 mapping were of high quality. 

9. Another informative diagnostic chart could be PCA plots of (1) all transcripts, (2) hypervariable discarded transcripts, and (3) retained transcripts.

The PCA plots that we present in (now) figure 3 are intended to show the general relatedness between datasets, not pick up subtle internal signals, be they read count variation within few transcripts or more complex evolutionary relationships. Within each of the 44 datasets, given that there are 26,107 transcripts involved that harbour varying count values, from 0 to high, as well as varying levels of intra-condition count variation (across the “conditions” that we were interested in), we felt that although the PCA plots were appropriate for demonstrating the very general evolutionary based relationship between datasets, they may not have the resolution at this per dataset level, to pick up the subtle shift in signal of removing the relatively few transcripts with highest levels of intra condition noise. At the reviewers request we have generated the corresponding PCA plots (Fig. R1 embedded in uploaded reviewer response file), minus the transcripts harbouring high levels of intra-condition noise, but visually these do not change much from the original PCA’s included within the paper. We have included these plots here, but have not done so within the manuscript. We have clarified within the methods that we are using PCA plots to summarize the more generalized relationships between datasets (line 336).

<image is in the response_to_reviewers.docx uploaded file>

Figure R1. PCA plots of datasets following the removal of intra-condition variation. PCA plots based on normalized non-filtered count data of the individual datasets comparing wolf and dog (top), and tame and aggressive fox (bottom) samples after the removal of transcripts harboring levels of intra condition variation above the the 95th and 97th percentile thresholds respectively. In the latter only individual samples that were positioned within a distant cluster are labelled with the sample ID. 

10. This does not sound right. “The number of transcripts removed was higher for the fox samples than for those of wolf and dog, reflecting the higher intra-condition variability present.” If percentiles of transcripts are being discarded, shouldn’t the proportion of detected transcripts discarded be the same for both studies? It is not explained clearly.

We have clarified this in in the manuscript (line 224), and the reviewer was correct here in saying this is ambiguous. We use the variation value associated with a specific percentile as our cut-off. This does not necessary mean that the number of transcripts removed are the same each time. For example, within one comparison the variation value at the 95th percentile could be one number but within a different comparison the variation value at the same threshold could be different, as the overall distribution is dependent on the input datasets. This is actually one of the problems in explaining our exploration of the data that the now added simulations greatly helps with; as we can specify the level of background intro-condition variation explicitly and in so doing so the numbers removed within each iteration of the same level would be expected more similar (aside from some stochastic change based on the random nature of introduced variation).

11. The figures should be explained in sequence. Eg: Figure 4C and the minitable in Fig 4 should be explained in the text before Fig 5a.

We have now split the original figure 5 into to separate figures (Fig. 5 and Fig 7), and referred to them in the appropriate order (including that of the embedded table), as indicated by the reviewer. Note, the original figure and table numbers have been altered as a result of the addition of new figures and tables where required – but the order is not referred to correctly for all.

Reviewer #2: 

Lobo et al present an evaluation of their software TVscript, which evaluates intra-condition variability in the counts that have been mapped to a transcriptome in an RNA-Seq experiment and removes the transcripts associated with the highest level of this variability, up to a user specified percentile threshold. They test the software by applying it to two pairs of datasets from wild and tame animals, wolves vs dogs and aggressive vs tame foxes. The greatest fraction of differentially expressed transcripts (DETs) is obtained by removing 3 to 5% of transcripts, and the authors describe some interesting features of the gene families of the corresponding differentially expressed genes, including common changes upon taming. The approach to RNA-Seq analysis is a potentially interesting one, representing another approach similar in some ways to the “orthogonal filtering” of low-expressed transcripts that is commonly used to increase the power in the analysis of RNA-Seq experiments. Unfortunately there are a number of aspects to the methodology that make it hard to recommend publication in the present form.

We thank this reviewer for taking the time to review our manuscript and we have invested significant time and effort in responding to the individual comments provided. We hope that after reviewing the changes made, the reviewer will now agree that our manuscript has been sufficiently improved. In relation to the first comment, this overlapped with what reviewer #1 pointed out so there repetition in our response.

1 Most importantly, I don’t think the data sets analysed are appropriate for the main intention of the paper. It is hard to tell whether the alterations made to the transcriptome improve the results rather than inducing false positives. The data sets used for testing come from multiple batches, and two organisms, one of which is appreciably divergent from the transcriptome to which it is aligned. In short, there are too many other uncontrolled factors in the analysis done to tell whether the results are reliable. In testing TVscript, it would be better to use an approach like that taken in Rapaport et al 2013 (https://doi.org/10.1186/gb-2013-14-9-r95), which uses data sets where batch effects are better controlled, including one (GEO GSE 49712) where external rna control consortium (ERCC) spike ins were used to produce known true positive DEGs.

We now extensively explore the effects of intra-condition per-transcript read count variation on the identification of differentially expressed transcripts through a series of differential expression analysis experiments involving highly controlled simulated count datasets derived from the available dog reference transcriptome. These are mentioned within the introduction (line 130) and described in detail within the materials & methods section under the new section titled “Controlled intra-condition variation within simulated data” (starting at: line 247). Using simulated count datasets allowed the exact level of background intra-condition count variation to be specified, as well as the specification of a subset-set of transcripts to be over-expressed across replicates within second conditions used (for each experiment performed). Within our simulated replicates, background count variation refers to varying count levels associated with individual transcripts that are not maintained across replicates (of a given condition), thus effectively reflecting intra-condition noise. On the other hand specifying a subset of transcripts to be over represented across replicates of a condition reflects identifiable over expressed transcripts. 

Following a brief introductory set of differential expression experiments on simulated data where levels of intra-condition variation are kept at zero (line 265), a set of simulations where the levels of background variation explored went from 1% to 10% in steps of one were performed. At each increment one hundred iterations of the following were performed (line 282): (i) Ten replicates of count dataset were generated, representing reads evenly spread across 22,580 dog transcriptome transcripts (after normalization by length) ranging in length from 300 to 5000 bp, and allocated into two conditions A and B (five in each). Within B one hundred selected transcripts had counts over represented. (ii) At the percent level of variation associated with the iteration, that percent of transcripts from each of the ten replicates were randomly selected for count over-representation. (iii) DESeq2 was used on the count files within conditions A and B to obtain a list of over expressed transcripts. (iv) TVscript was run using a 95th percentile variance threshold to generate ten corresponding modified count files separated into two conditions (A’ and B’). (v) DESeq2 was again used on these to obtain a list of over-expressed transcripts. (vi) The lists of over-expressed transcripts obtained in (iv) and (v) were cross compared. This was repeated in two different ways: (i) the one hundred transcripts initially flagged for over representation were kept constant throughout all levels of variation and for each of the associated iterations and (ii) during each level of variation and for each iteration a new set of one hundred transcripts were randomly selected. 

The results are presented within two new paragraphs (staring at: line 414) and within the newly added figures (Fig. 2 and Fig. S5), as well as Tables S2, S3, S4 and S5. In brief, the results in the figures indicate that within each iteration of steps (i) to (vi) there are slightly more of the one hundred transcripts selected for read over representation (within condition B’s of each iteration) identified post filtering. Tables S2 and S3 provide the counts of commonly identified differentially expressed transcripts (before and after filtering) for each of the one hundred iterations, at each level of intra-condition variation introduced. However even if the numbers are high they must be taken in the context of the maximum identified from either pre- or post filtered data. This is why tables S4 and S5 were provided as they present the agreeability relative to these maximums, and from levels of randomly introduced intra-condition count variation going from 1 to 4% the agreeability is high (Table S4 - 1 to 4% averages: 0.96, 0.98, 0.94 and 0.81; Table S5 - 1 to 4% averages: 0.96, 0.97, 0.93 and 0.82) (line 447). Finally, within figure 2 and figure S5, at and above the 5% level of intra-condition variation the ability to successfully identify the one hundred transcripts selected for over representation within condition B greatly diminishes within iterations. This could be indicative of a tentative estimate on the limit of at what level of random intra-condition count variation becomes inhibitory within differential expression analysis studies (line 452).

We thank (both) reviewers for providing dataset suggestions with larger numbers of replicates than we initially used in our analysis and pointing out this area that could be improved. We felt that using simulated data for this newly included analysis allowed us to explicitly control all relevant factors involved and allowed us to recreate the many replicates under the varying degrees of intra-condition variation described. We feel that this comment has contributed greatly to the clarity of our exploration in relation to the effects of intra-condition read count variation on the detection of differentially expressed transcripts.

2 More detail is necessary about how the differential expression was performed; figure 2c seems to show that the dogs do separate by batch (1-5; 6; 7; 8-9) and one would normally use a design formula that took account of the different sources of the data, something like ~ batch + tameness.

(although the brain tissue and instrument used are also similar for dogs 6-9, so one could also try ~ tissue + tameness). The authors should state whether they used a formula like this, and justify why not if they did not (Ideally, the R script used for differential expression analysis could be made available).

We have now added the following text to the manuscript in relation to this comment (line 318):

“For the aggressive vs. tame fox case study batch effects were not considered as all data came from the same study, tissue and sequencing run, additionally no further information about sample preparation was available. For the wolves vs. dogs case study we tested for effects based on tissue, primarily for quality control of the final transcripts we drew biological-related conclusions about, and compared results obtained to those in the absence of batch information. In our analysis we used differential expression results based solely the latter, as firstly, effects associated with tissue at an inter-study level are unpredictable as there are many factors involved, such as precision of dissection, time of dissection, time to dissect, state of individual tissue samples as well as individual who prepared sample, and other than publication or information mentioned for the fox case study, no further information on batches was obtainable. Secondly, although DESeq2 provides an internalized method for accommodating batch effects that we applied (~batch + condition), the results obtained at an intra-study level, with well defined batches, between alternative methods of testing are variable [52]. Lastly, we were primarily exploring the effects of removing hyper variable transcripts on the mechanics of detecting differentially expressed transcripts and our simulations and case studies were a means to an end in achieving this. As long as input counts for a given filtering threshold within a given case study or a iteration were consistent with those of the initial input data, the effects of removing hyper variable transcripts could be observed, independent of other factors affecting the data prior to analysis.”

We have also now included the new supplementary figure S6 that confirms the present of six of the shared genes between dogs and tame foxes once batch effects based on tissue have been taken into account. This figure is referred to within the results section of the manuscript (lines 580 and 602).

For the newly added simulated testing framework (described for comment 1), we had strict control of all parameters within the simulated data within each of the differential expression analysis experiments that was performed. For example, we could select the level of random background count variation allowed within the other wise evenly distributed count values as well as select the transcripts to have counts over represented across multiple replicates of a given condition. Although such a scenario may be somewhat artificial in vivo, it allowed us to explicitly look at the effects of before filtering and after filtering within each iteration without having to deal with batch or other sources of variation.

I would also remark that, since the authors emphasise that data comes from different sources, it was not immediately clear, until I looked at supplementary table 1, that the wolves and dogs 1-5 all come from one study, and similarly all the foxes, aggressive and tame, come from one study. This should be brought out more in the text, as otherwise the reader is made to wonder how any difference between wild and tame will be detectable that is not confounded with batch effects.

We have now clarified this with the method section (line 166).

3 It is unclear to me why the authors used the C. familiaris transcriptome for their work on the fox as well as the dog/wolf, when the fox and wolf lineages diverged 10 myr ago. A genome and transcriptome are available for the fox (https://www.ensembl.org/Vulpes_vulpes/Info/Index?db=core), and even though it is of lower quality than the dog, a higher mapping rate might have been expected. I appreciate that it makes the assessment of TVscript, and to some extent the comparison of dog and fox DET results, more straightforward (though information on orthology is also available). The low mapping rate and slightly strange clustering of the points in the PCA plot fig 2d are indications there may be some problems with the fox data that might in part come from the choice of transcriptome, and this casts some doubt on the DET results for me.

In relation to the lower fox mapping numbers within the materials & methods we now map all of the fox data to the available fox transcriptome (line 191). Mapping numbers that were achieved (new figure S3) when mapped are similar to that when these data were mapped to the dog transcriptome (line 410). Within the discussion we explicitly state that: 

“High intra-condition count variation at an inter, and to a lesser extent intra, study level can arise from a range of sources including i) biological differences between samples such as age, sex, diet, and health; ii) in silica error involving assembly tools producing poorly understood chimeras within the reference transcriptome [50,60,61]; iii) ambiguities in read mapping to such references [62]; iv) normalization of count data derived from such mapped reads [63]; and v) including in vitro error during library preparation protocols [64,65].”

The main reason why we selected the dog transcriptome for our analysis was that we wanted to use a common reference set between the dogs vs. wolves and aggressive vs. tame foxes comparisons, and we believed that the dog reference transcriptome, being more commonly worked on, is more refined and possibly less vulnerable to internal ambiguities such as chimeras, partial redundancy and missing transcripts. Given our primary goal was to explore the effects of per-transcript intra-conditon count variation on the detection of differentially expressed transcripts, we felt that the dog reference transcriptome was a good place to start. We realise that if our main approach was to solely (or primrily) study differential expression alterations during domestication, information on orthology is available; but this was more of a side product for us to make our exploration on intra-condition count variation more interesting. It is likely because of this ballance between testing and biological application that the reviewer suggests two different papers in comment 4; and we do understand the point being made.

More generally in terms of the mapping approach used, we discuss this in relation to comment 7 made by reviewer #1, where we confirm numbers using the alternative kallisto based approach and provide all counts obtained following mapping by Bowtie2 and kallisto. In all cases these correlations between are exceptionally high and we have discussed this within the results section of the manuscript (line 401) and included a new supplementary figure (Fig. S2). All read counts and abundances by Bowtie2 and kallisto have now been made available on the Zenodo repository (line 402) (https://zenodo.org/record/6778429).

Generally the reviewer comment “… are indications there may be some problems with the fox data that might in part come from the choice of transcriptome, and this casts some doubt on the DET results for me …” is true, but this is also true for differential expression analysis studies where there is not an established high quality, closely related, reference available, or when results are dependent on de novo assembled contigs where there can be many ambiguities, or indeed when single cell sequencing has not been performed. There should usually be some level of doubt. This is one of the reason why we emphasis that this study is an exploration of the effects of intra-condition count variation on the detection of differentially expressed transcripts (largely independent of reference set used), and why the identification of the genes commonly over or under expressed within dogs and tame foxes are providing “tentative” support previously identified genes, but not definitive proof. At the end of our concluding paragraph we have added the following sentences to highlight this (line 730):

“We use the word tentative to describe our support as the primary aim of this study was to investigate the effects of intra-condition count variation on the detection of differentially expressed transcripts, and the identification of genes involved within an evolutionary process, such as domestication, should be supported by datasets specifically generated for that purpose, and confirmed relative to the different reference transcriptomes involved. The quality of such transcriptomes in turn, in relation to chimeras, missing transcripts and partial redundancies, must also be carefully explored.”

My recommendation would be to split the work into two papers, one comparing the wild and tame animals, which to me was the most interesting part of the manuscript, and one assessing TVscript. It appears to me from comparing Tables 1 and 2 in the paper (filtered transcriptome) with supplementary table 6 (unfiltered transcriptome), that the filtering did not make a very big difference here. Hence the first paper could use the results from the more standard methods of supp. Table 6 and the interesting overlaps between the changes on domestication in the two pairs of animals would still be largely maintained. The second paper really would really need to use different test data sets , as suggested above, to establish whether TVscript is genuinely increasing sensitivity without introducing type I errors.

The extensive testing on simulated data indicates that the (small) difference TVscript makes is consistent across iterations, involving varying levels of introduced intra-condition variation. It is this point that we are trying to highlight, i.e. that the identification of differentially expressed transcripts should be presented within the context of intra-condition count variation and the alterations observed as a result removing hypervariable transcripts can be important, independent of the normalization approached used by the utilized differential expression tool. Often it is one or few such transcripts that paper conclusions are based on. In relation to this we have added the following text to the concluding paragraph of the discussion (line 718):

“We propose that studies using RNA-seq data at an inter, or intra, study level should determine whether or not transcripts identified as being differentially expressed, using pre-filtered reference sets, are still identified once filtering based on intra-condition count variation as been performed; regardless of the differential expression software used (or the method of obtaining initial counts). Discussion of such transcript can then be taken into the context of ambiguity observed. Such context is likely going to be dataset specific, as indicated between differences between our case studies, as the extent of intra-condition count variation will differ between datasets and will rarely be known as a prior to analysis.”

More generally, this study has been on-going for some time and the reviewer has highlighted something that we have struggled with since the start. Initially, the TVscript manuscript was going to be an application note, focussed mainly on testing, then it became more about a literature review/domestication study using published data and at one point we were attempting to generate our own dog and wolf transcriptomic data to include. But this got set-aside for various reasons. The eventual manuscript that emerged was (we feel) an interesting hybrid, but more focused on the software. As it stands we have come up with a relevant piece of work, especially following responses to both reviewers comments, that encompasses a reasonable compromise that should be of implicit interest within the field of transcriptomics. We feel that the manuscript adds relevant information to an important topic that should be considered more widely when discussing identified differentially expressed transcripts.

I would like to thank the authors for preparing the manuscript carefully, providing detailed results and supplementary material, and providing access to the code of TVScript along with links to other useful material on sourceforge.

Once again we thank the reviewer for their time and effort. We are glad that they liked the presentation of our work, and more generally, our background resources. We plan to produce more. The comments provided here have added greatly to our manuscript.

---

## [Decision Letter · Decision Letter 1]

10 Aug 2022

PONE-D-21-32424R1On taming the effect of transcript level intra-condition count variation during differential expression analysis: a story of dogs, foxes and wolvesPLOS ONE

Dear Dr. Archer,

Thank you for submitting your manuscript to PLOS ONE. After careful consideration, we feel that it has merit but does not fully meet PLOS ONE’s publication criteria as it currently stands. Therefore, we invite you to submit a revised version of the manuscript that addresses the points raised during the review process.

Both reviewers are overall very happy with your revised version. However, they both have some minor comments that require final clarification. I don't anticipate these points will take too much time to address and look forward to reading the final version.

We look forward to receiving your revised manuscript.

Kind regards,

Katherine James, Ph.D.

Academic Editor

PLOS ONE

Journal Requirements:

Reviewers' comments:

Reviewer's Responses to Questions

**Comments to the Author**

1. If the authors have adequately addressed your comments raised in a previous round of review and you feel that this manuscript is now acceptable for publication, you may indicate that here to bypass the “Comments to the Author” section, enter your conflict of interest statement in the “Confidential to Editor” section, and submit your "Accept" recommendation.

Reviewer #1: All comments have been addressed

Reviewer #2: (No Response)

2. Is the manuscript technically sound, and do the data support the conclusions?

Reviewer #1: Yes

Reviewer #2: Yes

3. Has the statistical analysis been performed appropriately and rigorously? 

Reviewer #1: Yes

Reviewer #2: I Don't Know

4. Have the authors made all data underlying the findings in their manuscript fully available?

Reviewer #1: Yes

Reviewer #2: Yes

5. Is the manuscript presented in an intelligible fashion and written in standard English?

Reviewer #1: Yes

Reviewer #2: Yes

6. Review Comments to the Author

Reviewer #1: I commend the authors for the comprehensive amendments and explanations. I think the article is in great shape. The provision of scripts and data on zenodo is appreciated. Please consider the following points as optional suggestions.

1. This new simulation is a welcome addition that supports the perceived need for a tool like TVscript.In figure 2, I would recommend putting the legend for light and dark grey boxes on the plot itself.

2. OK

3. OK

4. OK

5. OK, but the passage on line 723 should be written in a clearer, more straightforward way.

6. OK.

7. OK, but "high r2 correlation values" should be qualified with a specific range (eg: 85-98%), so that the reader can understand what "high" means.

8. OK

9. OK.

10. OK

Typo: "we also had an interested in understanding whether"

Reviewer #2: I am grateful to Lobo et al for their efforts in addressing my criticisms of the first version of their manuscript.

1 Appropriateness of data sets used to assessing TVscript: I think that by using extensive simulated data the assessment of the behaviour of TVscript is much improved.

2 more detail has been given on how the DE analysis was performed as requested. I am not totally convinced by the lengthy discussion of unknown effects (though there is no reason to remove it); clearly there are always unknown factors, but that does not affect the apparent effect of batch in the PCA plots. But in any case, the important test, that including batch in the statistical model for DE in the dogs, has been done and the results (fig S6) seem to confirm that it does not have a very large effect

3. I thank the authors for checking the mapping rate of the fox data to the fox transcriptome (and, incidentally, by mapping with kallisto as well). Even though it turned out not to affect the mapping rate very much, I feel this was an important check to perform.

4. I appreciate that factors outside the authors’ control can affect the way that a project ends up being carried out and written up. Although my suggestion was to divide the manuscript, I do not insist on it, I am content with the manuscript’s current form.

The criticisms that I made in my first review, at least, are allayed. However the first reviewer raised other serious points particularly point 5 about filtering where the sample group information is used (see Bourgon, Gentleman, and Huber 2010). I did not spot this in my own review and it is for the first reviewer to assess whether the additional simulations have addressed this satisfactorily. I did notice that in the discussion of the matter in the DESeq2 vignette (http://bioconductor.org/packages/devel/bioc/vignettes/DESeq2/inst/doc/DESeq2.html#independent-filtering-and-multiple-testing) a histogram of the p-value of the filtered genes is provided, showing that it is approximately uniform. I would tentatively suggest (again, I defer to the first reviewer here ) that, done for transcripts rather than genes, a p-value histogram could provide an empirical way of demonstrating that the filtering is independent of the test statistic under the null hypothesis, if required.

7. PLOS authors have the option to publish the peer review history of their article (what does this mean?). If published, this will include your full peer review and any attached files.

Reviewer #1: No

Reviewer #2: No

---

## [Author Response · Author response to Decision Letter 1]

15 Aug 2022

Reviewer #1

“I commend the authors for the comprehensive amendments and explanations. I think the article is in great shape. The provision of scripts and data on zenodo is appreciated. Please consider the following points as optional suggestions.”

We thank the reviewer for these minor revisions and we have adjusted the manuscript accordingly. The time and effort that the reviewer has spent on this, and on the previous, round of review has been invaluable for our manuscript.

“1. This new simulation is a welcome addition that supports the perceived need for a tool like TVscript.In figure 2, I would recommend putting the legend for light and dark grey boxes on the plot itself.”

We have now added a key to figure 2 explaining the light and dark grey boxes. This has also been added to the similar figure S5.

“2. OK

3. OK

4. OK

5. OK, but the passage on line 723 should be written in a clearer, more straightforward way.”

We have reworded the three lines following line 723.

“6. OK.

7. OK, but "high r2 correlation values" should be qualified with a specific range (eg: 85-98%), so that the reader can understand what "high" means.”

We have now added the range to the text (on line 405) by saying: “R2 values ranged between 0.8546 and 0.9944.”

“8. OK

9. OK.

10. OK

Typo: "we also had an interested in understanding whether"”

Corrected

Reviewer #2 

“I am grateful to Lobo et al for their efforts in addressing my criticisms of the first version of their manuscript.”

We are very grateful for the comments provided by this reviewer that, in conjunction to those provided by the first reviewer, has greatly clarified and improved our work.

“1 Appropriateness of data sets used to assessing TVscript: I think that by using extensive simulated data the assessment of the behaviour of TVscript is much improved.”

Agreed. The simulations allowed us to test our approach within a highly controlled framework prior to application on the real data.

“2 more detail has been given on how the DE analysis was performed as requested. I am not totally convinced by the lengthy discussion of unknown effects (though there is no reason to remove it); clearly there are always unknown factors, but that does not affect the apparent effect of batch in the PCA plots. But in any case, the important test, that including batch in the statistical model for DE in the dogs, has been done and the results (fig S6) seem to confirm that it does not have a very large effect”

Testing for batch effects was an important addition that we had not initially included. We feel that the inclusion of this, along with the results based on simulations (where batch effects less relevant), has greatly improved completion when presenting our results.

“3. I thank the authors for checking the mapping rate of the fox data to the fox transcriptome (and, incidentally, by mapping with kallisto as well). Even though it turned out not to affect the mapping rate very much, I feel this was an important check to perform.”

This was an important check that we should have provided within the first version in order to set the mind of readers at rest relative to the question raised by the reviewer. We feel that it is an interesting point and we are pleased to have made this addition at the reviewer’s suggestion.

“4. I appreciate that factors outside the authors’ control can affect the way that a project ends up being carried out and written up. Although my suggestion was to divide the manuscript, I do not insist on it, I am content with the manuscript’s current form. The criticisms that I made in my first review, at least, are allayed.”

Great, we completely understood why the reviewer was thinking about two separated papers here, and we will still aim to produce a future paper with our own generated RNA-seq datasets.

“However the first reviewer raised other serious points particularly point 5 about filtering where the sample group information is used (see Bourgon, Gentleman, and Huber 2010). I did not spot this in my own review and it is for the first reviewer to assess whether the additional simulations have addressed this satisfactorily. I did notice that in the discussion of the matter in the DESeq2 vignette (http://bioconductor.org/packages/devel/bioc/vignettes/DESeq2/inst/doc/DESeq2.html#independent-filtering-and-multiple-testing) a histogram of the p-value of the filtered genes is provided, showing that it is approximately uniform. I would tentatively suggest (again, I defer to the first reviewer here) that, done for transcripts rather than genes, a p-value histogram could provide an empirical way of demonstrating that the filtering is independent of the test statistic under the null hypothesis, if required.”

Reviewer 1 was happy with our response to their point 5 and did not require any further clarification. Given that reviewer 2 has stated that “… I defer to the first reviewer here …” we feel that point 5 has been sufficiently discussed within the manuscript for the scope of our study.

---

## [Editor Report · Decision Letter 2]

1 Sep 2022

On taming the effect of transcript level intra-condition count variation during differential expression analysis: a story of dogs, foxes and wolves

PONE-D-21-32424R2

Dear Dr. Archer,

We’re pleased to inform you that your manuscript has been judged scientifically suitable for publication and will be formally accepted for publication once it meets all outstanding technical requirements.

Kind regards,

Katherine James, Ph.D.

Academic Editor

PLOS ONE
---

## [Editor Report · Acceptance letter]

12 Sep 2022

PONE-D-21-32424R2 

On taming the effect of transcript level intra-condition count variation during differential expression analysis: a story of dogs, foxes and wolves 

Dear Dr. Archer:

I'm pleased to inform you that your manuscript has been deemed suitable for publication in PLOS ONE. Congratulations! Your manuscript is now with our production department. 

Kind regards, 

on behalf of

Dr. Katherine James 

Academic Editor

PLOS ONE